



# SHAFTS (v2022.3): a deep-learning-based Python package for Simultaneous extraction of building Height And FootprinT from Sentinel Imagery

Ruidong Li[1], Ting Sun[2], Fuqiang Tian[1], and Guang-Heng Ni[1]

[1]Department of Hydraulic Engineering, Tsinghua University, Beijing 100084, China
[2]Institute for Risk and Disaster Reduction, University College London, Gower Street, WC1E 6BT, UK

**Correspondence:** Ruidong Li (lrd19@mails.tsinghua.edu.cn)

**Abstract.** Building height and footprint are two fundamental urban morphological features required by urban climate modeling. Although some statistical methods have been proposed to estimate average building height and footprint from publicly available satellite imagery, they often involve tedious feature engineering, which is hard to achieve efficient knowledge discovery in a changing urban environment with ever-increasing earth observations. In this work, we develop a deep-learning-based

(DL) Python package–SHAFTS (Simultaneous building Height And FootprinT extraction from Sentinel Imagery) to extract such information. Multi-task DL (MTDL) models are proposed to automatically learn feature representation shared by building height and footprint prediction. Besides, we integrate Digital Elevation Model (DEM) information into developed models to inform models of terrain-induced effects on the backscattering displayed by Sentinel-1 imagery. We set conventional machine-learning-based (ML) models and Single-task DL (STDL) models as benchmarks and select 46 cities worldwide to evaluate

developed models' patch-level prediction skills and city-level spatial transferability at four resolutions (100 m, 250 m, 500 m and 1000 m). Patch-level results of 43 cities show that DL models successfully produce more discriminative feature representation and improve the coefficient of determination ($R^2$) of building height and footprint prediction over ML models by 0.27-0.63, 0.11-0.49, respectively. Moreover, stratified error assessment reveals that DL models effectively mitigate severe systematic underestimation of ML models in the high-value domain: for the 100 m case, DL models reduce Root Mean Square

Error (RMSE) of building height higher than 40 m and building footprint larger than 0.25 by 31 m and 0.1, respectively, which demonstrates the superiority of DL models on refined 3D building information extraction in highly urbanized area. For the evaluation of spatial transferability, when compared with an existing state-of-the-art product, DL models can achieve similar improvement on the overall performance and high-value prediction. Furthermore, within the DL family, comparison in building height prediction between STDL and MTDL models reveals that MTDL models achieve higher accuracy in all cases and

smaller bias uncertainty for the prediction in the high-value domain at the refined scale, which proves the effectiveness of multi-task learning on building height estimation.



## Special Terms

$\mathrm{RMSE}'$ centered RMSE. 23

$\mathrm{R}^2$ Coefficient of Determination. 1, 16, 17, 22–24

$\mathrm{wR}^2$ weighted $\mathrm{R}^2$. 16, 17

**CC** correlation coefficient. 16–18, 23, 25, 30

**CCAP** the city clustering algorithm based on percolation theory. 30, 31, 36

**CNN** Convolutional Neural Networks. 5, 11–15

**DMP** Differential Morphological Profile. 35

**FC** Fully-Connected. 12

**GAP** Global Average Pooling. 12

**GLCM** Gray-Level Co-occurrence Matrix. 35

**IQR** Interquartile Range. 10, 32

**MAE** Mean Absolute Error. 16, 17

**ME** Mean Error. 16–20, 23

**MTDL** Multi-task DL. 1, 5, 6, 14, 16, 19, 25–27, 29–34

**MTL** Multi-Task Learning. 12–16, 25

**NIR** Near Infra-Red. 7, 15, 34, 35

**NMAD** Normalized Median Absolute Deviation. 16–18, 20–22

**RFR** Random Forest Regression. 4, 5, 16, 28–30, 36, 37

**RMSE** Root Mean Square Error. 1, 16–18, 20–24, 30, 31, 34

**SAR** Synthetic Aperture Radar. 7

**SHAFTS** Simultaneous extraction of building Height And FootprinT from Sentinel Imagery. 33, 34, 37



**STDL** Single-task DL. 1, 14, 16, 19, 25–27, 30–34

**STL** Single-Task Learning. 12, 14, 15

**SVR** Supporting Vector Regression. 4, 5, 16, 36, 37

**wRMSE** weighted RMSE. 16, 17

**XGBoostR** XGBoost Regression. 5, 36, 37





## 1 Introduction

Recent decades have witnessed a dramatic trend of global urbanization leading to significant transformation in the vertical structure and horizontal form of the Earth's landscapes (Frolking et al., 2013). Disturbance of the environment poses a substantial challenge to sustainable development, mainly through changes in land use and cover, biogeochemical cycles, climate, hydro-systems, and biodiversity (Grimm et al., 2008; Soergel et al., 2021). To meet the challenges, the fundamental information on urban morphology - particularly building footprint and height - is urgently needed for numeric modeling (e.g., flood

inundation (Bruwier et al., 2020), surface energy balance (Yu et al., 2021)) and policy formulation (Zhu et al., 2019b).

However, such information is only available in a limited number of cities based on diverse retrieval methods such as LiDAR measurement (Yu et al., 2010) and conversion from the number of floors observed in the street view images (Zheng et al., 2017), which hinders addressing the above challenges at larger scales in a consistent way. In particular, the emerging global models and related initiatives underline the necessity to develop consistent methods for generating 3D urban information worldwide

(Masson et al., 2020a; Sampson et al., 2015). Thus, a more consistent approach that can be applied at larger scales is highly desirable for extracting such information.

Thanks to the increasingly abundant and publicly available satellite imagery, 3D building information mapping from remote sensing images has received significant attention and achieved considerable advances (Esch et al., 2022). Moreover, the boosting of machine learning techniques has dramatically facilitated such mapping tasks (Li et al., 2020). According to target

granularity, mapping tasks can be categorized into two classes:

- The pixel-level tasks: target variables are predicted for each input pixel; such tasks often aim to seek a refined representation of individual buildings through very-high-resolution (VHR) images (Cao and Huang, 2021; Shi et al., 2020).

- The scene-level tasks: target variables are estimated on predefined spatial processing units; they produce aggregated properties of building groups at the scale of interests (Frantz et al., 2021; Geiß et al., 2020).

While pixel-level mapping gains its popularity in cases such as 3D city visualization, it often requires VHR images that are less publicly accessible. By contrast, scene-level mapping only requires medium-resolution (i.e., 10-100 m) images and suffices the need of urban land surface modeling for building characteristics at the neighborhood scale (∼ 500 m) (Mirzaei and Haghighat, 2010). Here, we focus on the scene-level extraction of 3D building information using a data-driven approach.

Being demonstrated great potentials in the scene-level extraction, the current data-driven approaches based on conventional

ML models (e.g., Random Forest Regression (RFR) (Li et al., 2020), Supporting Vector Regression (SVR) (Frantz et al., 2021) and ensemble regression (Geiß et al., 2020), etc.) have two major limitations:

- Over-reliance on manual data representation (Bengio et al., 2013). Current ML pipelines often need a feature engineering module that consists of carefully designed data preprocessing and transformation for feature calculation and selection (Frantz et al., 2021; Geiß et al., 2020). Although it provides an effective way for embedding a priori knowledge into

statistical models, it may also require a considerable amount of engineering skills and domain-specific expertise (LeCun et al., 2015). Moreover, since changing urban environment manifests complex spatial-temporal dynamics and emerging





multi-modal data add more insights to these intricate structures, fixed and manual representation lacks the flexibility to uncover valuable patterns and discriminative information from rich earth observations.

– Isolation in extraction tasks between building footprint and height. Although previous studies show that there exists no
strong correlation between building height and footprint (Li et al., 2020), it is recently found that the weak correlation may be attributed to the semantic gaps among buildings in various scenes (e.g., urban villages and peri-urban high-rises) (Guo et al., 2021) rather than the inherent nature of extraction tasks. In fact, the isolation in extraction tasks may confuse models by large intra-class differences of buildings and thus hinder them from learning good data representations.

To overcome the above limitations, we propose a new MTDL approach to facilitate 3D building information extraction at a
large scale with emphases on the following aspects:

- To reduce the reliance of ML models on manual data representation, we employ Convolutional Neural Networks (CNN), a DL model for its powerful representation learning ability in automatically and effectively extracting features from images via a composition of simple but non-linear modules (LeCun et al., 2015).

- To consolidate the extraction tasks between building footprint and height, we train the DL model in a multi-task way: both
the building footprint and height are set as optimization targets, so the training can benefit from the enriched representations of multi-dimensional information of buildings (i.e., footprint and height) by disentangled underlying factors and thus can improve the representation learning ability of models (Bengio et al., 2013). Furthermore, the learned representations can be shared among tasks to formulate a more lightweight model with significantly fewer parameters.

Besides, we aim to thoroughly evaluate the performance of the proposed MTDL approach in the scene-level 3D building
mapping. Although DL models have set new benchmarks on several remote sensing image processing tasks (e.g., land use classification (Zhong et al., 2020), building segmentation (Shi et al., 2020), local climate zones classification (Zhu et al., 2022), etc.), whether DL methods outperform ML ones on the scene-level 3D building mapping has yet to be examined.

Main procedures of the proposed workflow for 3D building information mapping include (Fig. 1):

– Dataset preparation. Reference cadastral data are collected from public sources and then rasterized into building height
and footprint maps by fishnet analysis. Target grids of the fishnet layer are filtered by some sample selection criteria and further retrieved as geocoded ground truth. According to target geolocation $(x_i, y_i)$ and timestamp $t_i$, Sentinel-1/2 imagery and SRTM data are selected from Google Earth Engine (GEE) (Gorelick et al., 2017) and then compose multi-band patches as corresponding explanatory variables.

– Model development. DL models utilize two input branches to directly ingest stacked band information from Sentinel-1/2
imagery and SRTM data. ML models including RFR, SVR and XGBoost Regression (XGBoostR) are also developed as baselines for comparison with CNN. For ML models, a feature calculation module is designed to incorporate single band statistics, multi-spectral indices, morphological characteristics and texture metrics to characterize the built environment (Frantz et al., 2021; Geiß et al., 2020).





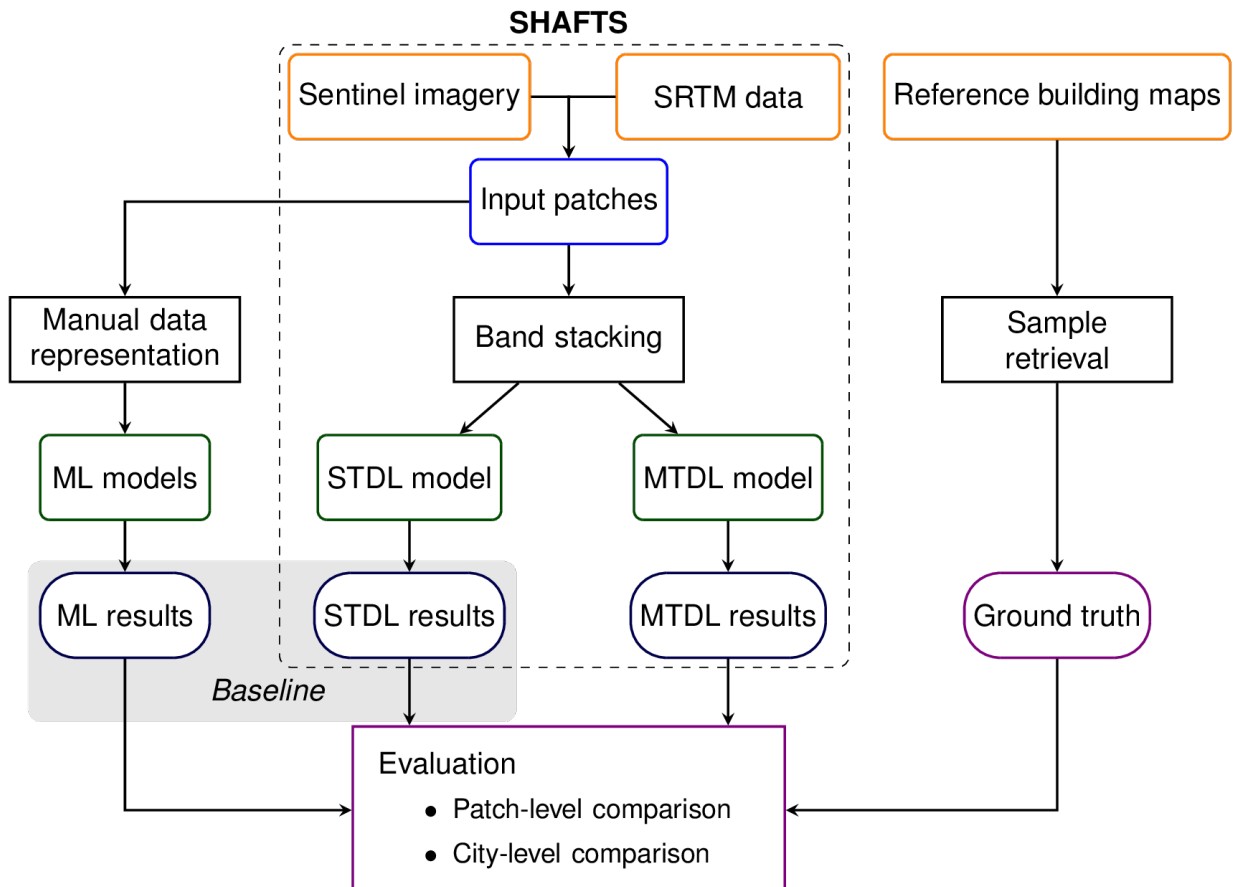

**Figure 1.** Overview of workflows to estimate building height and building footprint using ML and DL approaches.

- Performance evaluation. Specifically, we examine the prediction skills and spatial transferability of developed mod-
  els. In this work, prediction skills are evaluated by patch-level comparison in pattern statistics between reference and
  predictions, while spatial transferability is assessed by city-level comparison of 3D building structures mapped in test
  cities.

In the remainder of this paper, we first describe the proposed MTDL approaches for 3D building information extraction
(Sect. 2) and then perform diagnostic evaluation of model performance in 46 cities worldwide at both patch and city levels
(Sect. 3).





## 2 Model development

### 2.1 Theoretical background

For building footprint prediction, optical images are straightforward choices of inputs for most practices (Guo et al., 2021). Also, building height can be estimated from temporal patterns of building shadows which can be detected from time series of

optical images (Frantz et al., 2021). However, the quality of optical images is strongly influenced by weather conditions, which leads to research concerning alternative data sources for 3D building information retrieval. Synthetic Aperture Radar (SAR) satellite is one of such choices which is capable of all-weather day-and-night imaging (Moreira et al., 2013). SAR images are often displayed in the form of backscattering intensity which heavily depends on the material types, regional distribution and structural shapes of sensed objects (Koppel et al., 2017) and therefore is of great value to 3D building information retrieval. In

general, building height estimation from SAR images can be formulated a problem of inverse modeling: given the radar viewing geometry, underlying 3D building distribution is expected to be inferred from the observed backscattering results. Although radiometric analysis based on SAR models can help estimate building height in a simulating and matching way (Brunner et al., 2010), complex backscattering mechanisms and unknown prior knowledge about objects such as roof properties and building location constrain the model-based analysis in a real urban context (Sun et al., 2022). Confronted these challenges, this work

proposes to use deep neural networks to learn the solution of the inverse problem in a data-driven fashion. Furthermore, considering aforementioned prior knowledge can be derived from optical images and building footprint prediction, this work makes synergistic use of both optical and SAR images and estimates building footprint and height simultaneously.

Additionally, it is worthy of noting that in order to generate backscattering images for downstream quantitative studies, SAR data processing often requires geometric and radiometric correction on raw images acquired by sensors using a Digital

Elevation Model (DEM) (Loew and Mauser, 2007). Thus, the information of DEM can be critical for building height retrieval based on SAR images, especially for area with variable topography. Although some toolboxes–such as the Sentinel application platform (SNAP)–offer several processing functions to perform such corrections (Piantanida et al., 2021), this work proposes to include the information of DEM as an auxiliary explanatory variable in developed models to envelope terrain-induced effects on backscattering and compensate for possible limitations of existing algorithms.

### 145 2.2 Dataset preparation

#### 2.2.1 Explanatory data

Based on the aforementioned theoretical analysis and review of previous practices (Li et al., 2020; Frantz et al., 2021), this work selects the following explanatory data for 3D building information mapping:

– 10 m VV and VH polarizations of Ground Range Detected (GRD) Level-1 products of Sentinel-1 imagery (Torres et al.,
150    2012).

– 10 m red, green, blue and the near infra-red (NIR) band of Level-2A products of Sentinel-2 (Drusch et al., 2012).





 – 30 m DEM data from SRTM V3 product.

To construct a convenient engineering pipeline, this work implements functions to download satellite data from GEE through its Python interface, which includes the following steps:

155 1. Sentinel-1 GRD data acquired in Interferometric Wide-swath (IW) Mode, Sentinel-2 Level-2A data with cloud coverage < 20 % and SRTM data are retrieved from image collections stored in GEE.

   2. All images are filtered by constraints defined by spatial extent and record year derived from corresponding reference data.

   3. All bands are aggregated to annual medians for further model development.

160 It should be pointed out that unlike previous studies (Li et al., 2020; Esch et al., 2022), we do not prescribe urban boundaries as necessary inputs because we deem that such information is inherently encoded in the mapping results and can be extracted automatically by post-processing (Sect. 3.3.2).

### 2.2.2 Reference data

Based on previous research (Li et al., 2020; Cao and Huang, 2021) and additional search to our best efforts, we constructed a
165 dataset for 3D building information mapping, which consists of 46 cities worldwide (See Sect. SM1 in supplementary materials for access details). Most reference data used in this work are collected from publicly available sources provided by local governments and ArcGIS Hub (https://hub.arcgis.com/) to allow open research (Nosek et al., 2015). However, considering the inherent noises in open-source datasets due to geocoordinate offset, measurement error and misreporting (Burke et al., 2021), high-quality cadastral data of London and Glasgow prepared by Digimap (https://digimap.edina.ac.uk/) are also included in
170 this work. All reference data are saved as ESRI shapefiles so the building footprint can be calculated from the geometry area and height value can be queried from the attribute table, facilitating the subsequent multi-task learning of building footprint and height. We note that although some reference datasets (e.g., Beijing, Chicago, etc.) have available information of building floor numbers, the building height is yet to be obtained via conversion with assumed storey heights (Ji and Tang, 2020; Li et al., 2020) that may inevitably introduce bias in building height. Thus, we exclude them from model development but will use them
175 in the later model assessment (Sect. 3.3.1).

 The urban surface features high heterogeneity and complex process interactions at various scales (Masson et al., 2020b; Salvadore et al., 2015), which requires researchers to spatially aggregate the per-pixel data at an appropriate scale regarding specific downstream applications (Zhu et al., 2019b). To reveal 3D building characteristics at different scales (100 m, 250 m, 500m and 1000 m), average building height $H_{\mathrm{ave}}$ and building footprint $\lambda_p$ (also known as the plan area index) are calculated
180 by fishnet analysis with corresponding target resolutions:

$$\lambda_p = \frac{\sum_{i \in \mathcal{I}} A_i}{r^2} \tag{1}$$





$$H_{\mathrm{ave}} = \frac{\sum_{i \in \mathcal{I}} A_i \cdot h_i}{\sum_{i \in \mathcal{I}} A_i} \tag{2}$$

where $\mathcal{I}$ is the index set of buildings intersecting with the grid, $A_i$ is the intersection area of the $i$-th building and the pixel, $h_i$ is the height of the $i$-th building and $r$ is the target resolution which is equal to the grid spacing of the fishnet layer.

### 185    2.2.3    Data preprocessing

For scene-level prediction tasks, input satellite images are partitioned into patches using a moving window centered on the target pixel of the reference data, which facilitates models to ingest the information of surrounding environment. The window moves with a stride equal to the target resolution. Each patch is assigned with corresponding building footprint and height value as prediction targets. Thus, raw dataset $\mathcal{D}$ for building footprint and height prediction can be constructed as:

$$\mathcal{D} = \left\{ (\mathbf{X}_i, \lambda_{p_i}, H_{\mathrm{ave}i}) \mid \mathbf{X}_i \in \mathbb{R}^{s \times s \times 7}, i = 1, ..., N \right\} \tag{3}$$

where $s$ is the patch size (cf. Table. 1), 7 is the number of bands used in this work and $N$ is the number of samples.

Based on preliminary analysis of data records from reference datasets as well as checking rasterized results after fishnet analysis, we design several filtering rules for excluding noisy samples from model development:

1. The average building height value should fall into a reasonable interval. According to existing records considering the
geolocation of samples, this interval is predefined by: $2.0\,\mathrm{m} = H_{\mathrm{ave,min}} \leq H_{\mathrm{ave}i} \leq H_{\mathrm{ave,max}} = 500.0\,\mathrm{m}$.

2. The average building footprint value should be greater than a threshold $\lambda_{p_{\min}}$. Since footprint value can be calculated based on the summarization of binary classification results at pixel-level, we set the lower bound of model prediction as the area ratio between one pixel and the sample patch, i.e., $\lambda_{p,\min} = \frac{1}{s^2}$, with the assumption that deployed models have no access to the sub-grid information.

3. Grids with only few "sliver polygons" during fishnet analysis should be discarded for their inappropriate representation of building bulk properties (Lagacherie et al., 2010). According to Eq. 2, this case often gives rise to a sample patch with an extremely large building height but abnormally small footprint. In this work, we set the corresponding footprint threshold as $4\lambda_{p,\min}$ and the height threshold as 20 m which is close to twice the median of the whole dataset.

After sample selection, the whole dataset $\mathcal{D}$ is split into a training set $\mathcal{D}_{\mathrm{train}}$, a validation set $\mathcal{D}_{\mathrm{val}}$ and a test set $\mathcal{D}_{\mathrm{test}}$ using
two-step selection with different granularity. The first step is city-level selection. Three cities, Beijing, Chicago, Glasgow, are left out as the first part of the test set denoted as $\mathcal{D}_{\mathrm{test},1}$. The second step is patch-level selection. To be specific, patches from remaining 43 cities are randomly collected into $\mathcal{D}_{\mathrm{train}}$ (80%), $\mathcal{D}_{\mathrm{val}}$ (10%) and the second part of the test set denoted as $\mathcal{D}_{\mathrm{test},2}$ (10%) city by city. The patch-level sampling results under different target resolutions are summarized in Table 1.

In this work, both $\mathcal{D}_{\mathrm{train}}$ and $\mathcal{D}_{\mathrm{val}}$ are designed for model training. In regard to performance evaluation, $\mathcal{D}_{\mathrm{test},1}$ is prepared
for testing the spatial transferability of selected models because the surface information and satellite images of these cities are entirely unseen by models during training. $\mathcal{D}_{\mathrm{test},2}$ is designed to investigate the performance difference between ML





**Table 1.** Input sample patch size and number of samples under different target resolutions.

| Target resolution | 100 m | 250 m | 500 m | 1000 m |
|---|---|---|---|---|
| Input patch size $s$ | 200 m | 400 m | 800 m | 1600 m |
| Number of training samples $N_{\text{train}}$ | 1886810 | 478155 | 173582 | 60257 |
| Number of validation samples $N_{\text{val}}$ | 235853 | 59771 | 21696 | 7535 |
| Number of test samples $N_{\text{test},2}$ | 235828 | 59750 | 21674 | 7509 |
| Total | 2358491 | 597676 | 216952 | 75301 |

models and DL models and inform of better models for spatial transferability testing on $\mathcal{D}_{\text{test},1}$. We note that although a more rigorous procedure with $\mathcal{D}_{\text{val}}$ for hyperparameter finetuning may further optimize model performance, no further selection of hyperparameters is carried out as this work is not aimed to establish a new benchmark for certain datasets.

### 2.2.4   Dataset overview

The reference distributions of $H_{\text{ave}}$ and $\lambda_p$ of 46 cities in the reference dataset are shown in Fig. 2. According to reference distributions of $H_{\text{ave}}$ and $\lambda_p$, the majority of selected urban patches have the building footprint value ranging from 0.0 to 0.3 and height value ranging from 3 m to 40 m. Furthermore, the comparison of reference distributions under different target resolutions reveals negligible effects of spatial resolution on the typical distribution of $H_{\text{ave}}$ but strong influence on $\lambda_p$: when the target resolution decreases from 100 m to 1000 m, the median of $\lambda_p$ decreases from 0.13 to 0.008 while the median of $H_{\text{ave}}$ merely decreases from 8.7 m to 8.1 m. Also, the inter-quartile range (IQR) of $H_{\text{ave}}$ remains about 8.6 m under different resolutions while that of $\lambda_p$ decreases from 0.17 to 0.08. Such effect of spatial resolution indicates that surface heterogeneity is more vertically prominent: when spatial resolution comes close to the critical point satisfying homogeneous surface assumption with respect to $\lambda_p$, it would be still far from that of $H_{\text{ave}}$ so the distribution of $H_{\text{ave}}$ remains nearly unchanged.

We select five cities–Prince George (Canada), London (UK), San Francisco (USA), Chattanooga (USA) and Porirua (New Zealand)–that roughly lie on the boundary of urban clusters as representatives and present their Sentinel images in Fig. 2. They are located in different continents and exhibit various semantic categories: sparsely-built mid-risers in Porirua, open low-risers in Prince George, open high-risers in Chattanooga and compact low-risers in London and San Francisco. Based on visual interpretation, although the limited resolution of Sentinel-2's images might hinder us from identifying buildings in over-bright or shadow region with high confidence, most building footprints coincide well with corresponding polygons in reference building layers spatially, which demonstrates the reliability of reference datasets.



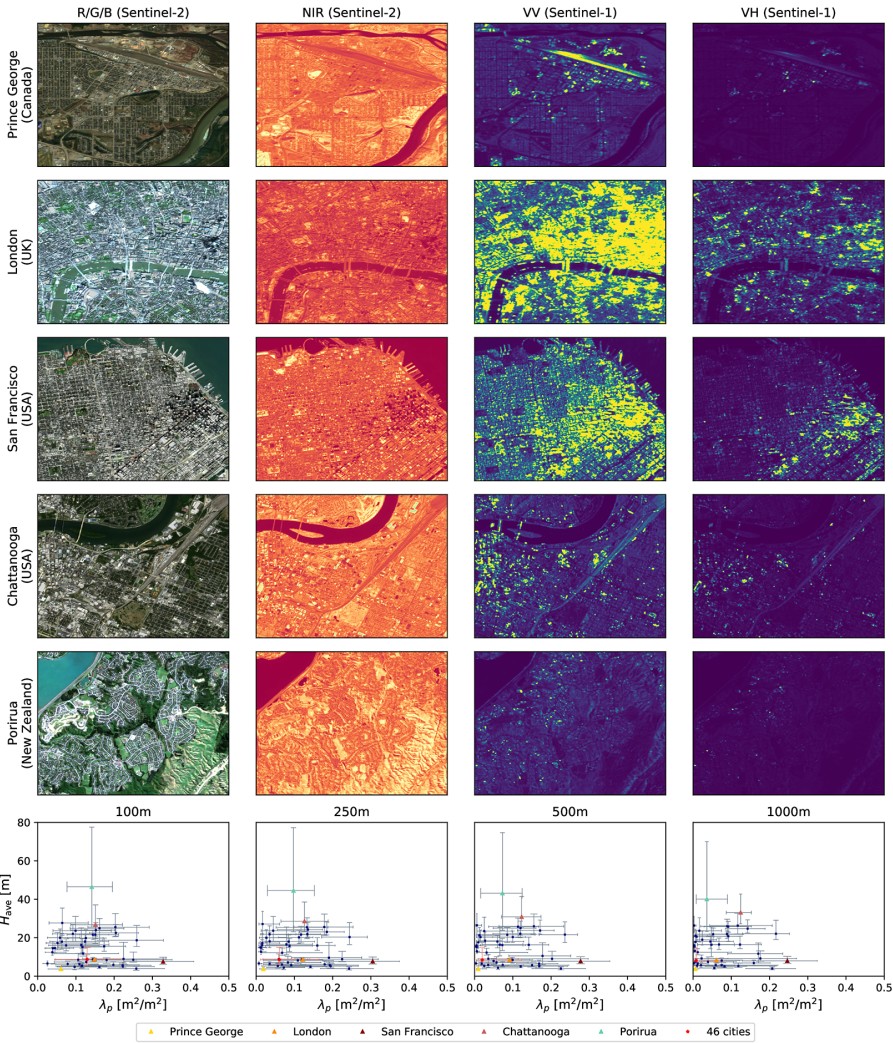

**Figure 2.** Top: Sentinel's images of five representative cities where band values are shown in their annual medians and reference building layers are overlaid on the R/G/B bands of Sentinel-2's images. Bottom: Reference $H_{\text{ave}}$ and $\lambda_p$'s distributions in 46 individual cities and overall dataset $\mathcal{D}$ where each pair of vertical and horizontal lines are characterized by upper, lower bounds and middle points which represent the quartiles of $H_{\text{ave}}$ and $\lambda_p$'s distributions, respectively.

## 2.3 Model description

### 2.3.1 Architecture of the CNN model

CNN models commonly consist of powerful backbones for hierarchical feature extraction and some heads for specific down-
stream tasks such as classification and regression. We adopt ResNet (He et al., 2016) integrated with Squeeze-and-Excitation



(SE) blocks (Hu et al., 2018) as our backbone model denoted as SENet (Fig. 3). Compared with prior architectures such as VGG (Simonyan and Zisserman, 2015) and Inception (Szegedy et al., 2015), ResNet adopts the framework of residual learning and inserts shortcut connections into the plain neural networks, which eases the training of much deeper models without degradation of performance (He et al., 2016). After the proposal of ResNet, various strategies have been introduced to further

improve its performance in computer vision tasks. One of the mainstream approaches is the visual attention mechanism which assigns adaptive importance weights to each channel or spatial position for higher emphasis on critical input elements (Qin et al., 2021; Zhu et al., 2019a). In this work, we use SE blocks within ResNet where channel-wise attention is calculated by two fully-connected (FC) layers after the global average pooling (GAP) on the input feature maps. In each SE block, the GAP produces an embedding of global spatial information for individual channels and the subsequent FC layers capture interde-

pendencies between channels based on the embedding so that per-channel modulation weights can be generated (Hu et al., 2018).

Furthermore, compared with standard practices in similar tasks using Sentinel imagery (Frantz et al., 2021; Geiß et al., 2020), we take the potential influence of local terrain relief on 3D building information mapping into consideration. Considering DEM is relatively a static dataset when compared with Sentinel imagery, we design another input branch of SRTM to allow the

potential capacity of our proposed CNN models in learning distinct temporal dynamics of different characteristics separately. After feature extraction is performed by individual backbones, two branches are merged by concatenating feature information from both Sentinel imagery and SRTM. Preliminary experiments show much improvement brought by the introduction of DEM information in building height prediction, especially at refined scales such as 100 m. See Sect. SM2 in supplementary materials for access details.

For simplicity, we use two FC layers in head design. The choice of final activation function depends on the target variable where the sigmoid function is used for building footprint prediction and Rectified Linear Unit (ReLU) function is used for building height prediction.

### 2.3.2 Multi-task learning

Multi-task learning (MTL) aims to improve learning for one task using the information enveloped in the training signals of

other related tasks (Caruana, 1997). Compared with single-task learning (STL), MTL features higher learning efficiency as well as less over-fitting risk since it guides models to reach more general feature representation preferred by multiple related tasks, which can be considered as an inductive bias for the regularization of deep neural networks (Ruder, 2017). One of common practices of MTL is to carry out multiple tasks simultaneously through the minimization of a weighted sum of individual losses (Eigen and Fergus, 2015). According to Fig. 3, we formulate the MTL loss $L$ as

$$L = w_1 \cdot \mathrm{Loss}_{\lambda_p} + w_2 \cdot \mathrm{Loss}_{H_{\mathrm{ave}}} \qquad (4)$$

where $\mathrm{Loss}_{\lambda_p}, \mathrm{Loss}_{H_{\mathrm{ave}}}$ are the loss functions of footprint and height prediction tasks, respectively.





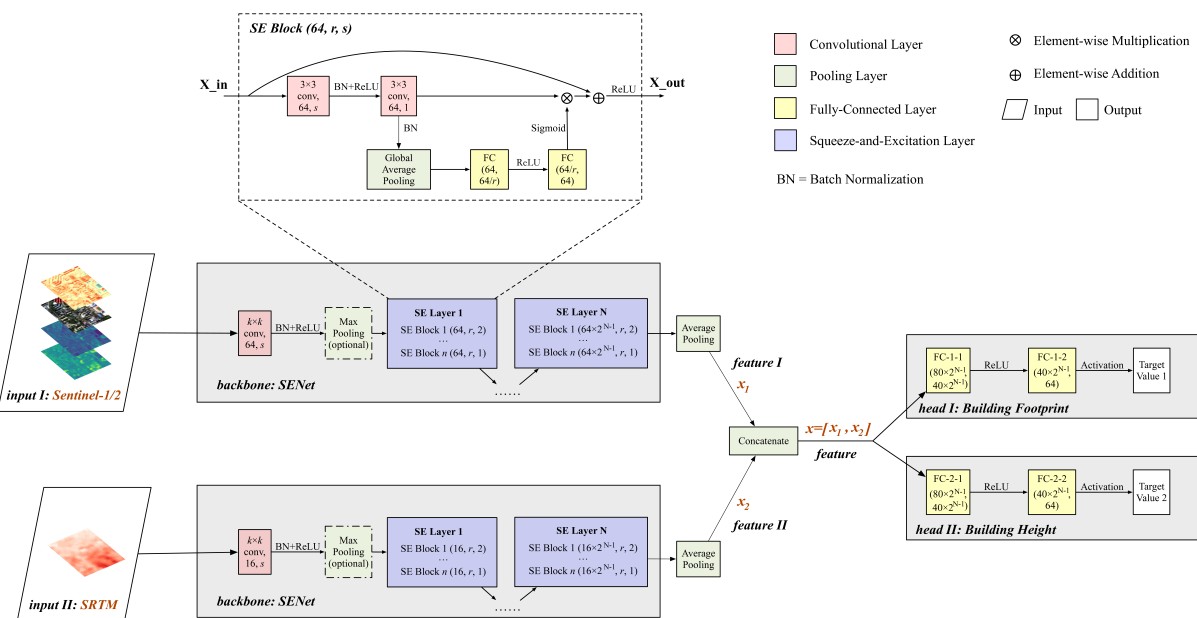

**Figure 3.** The architecture of the CNN deployed in this work. The convolutional layer is parameterized by $(k \times k$ conv, $c$, $s)$ where $k$ is the kernel size, $c$ is the number of output channels and $s$ is the stride. For initial convolution used in the backbone part, $k = 3, s = 1$ is chosen for the cases of 100 m and $k = 7, s = 2$ is chosen for the cases of 250 m, 500 m and 1000 m. The max-pooling layer is only used for the cases of 500 m and 1000 m. The fully-connected layer is parameterized by $(D_{\text{in}}, D_{\text{out}})$ where $D_{\text{in}}$ is the input dimension and $D_{\text{out}}$ is the output dimension. SE Block part is parameterized by $(C_{\text{in}}, r, s)$ where $C_{\text{in}}$ is the number of input channels, $r$ is a reduction ratio (Hu et al., 2018) for the fully-connected layer ($r = 16$ in this work) and $s$ is the stride of convolution. The number of SE Layers denoted by $N$ is chosen as 4 for the case of 1000 m and 3 for other cases (for the case of 100 m, the stride of the first SE Block in the first SE Layer is chosen as 1). The number of SE Blocks in each SE Layer denoted by $n$ is chosen as 1 for all cases.

While loss weights $w_1, w_2$ can be fixed as constants during the training, several dynamic weighting schemes have been proposed based on task uncertainty (Cipolla et al., 2018), gradient norm (Chen et al., 2018) or Pareto optimality (Sener and Koltun, 2018). In this work, we adopt two weighting schemes during model development:

- dynamic weight scheme based on task uncertainty (Cipolla et al., 2018; Liebel et al., 2020) where the loss function for MTL is reformulated as

$$L = \frac{1}{2\sigma_1^2} \cdot \text{Loss}_{\lambda_p} + \frac{1}{2\sigma_2^2} \cdot \text{Loss}_{H_{\text{ave}}} + \log \sigma_1 + \log \sigma_2 \tag{5}$$

where $\sigma_1^2$ and $\sigma_2^2$ are uncertainty measures for corresponding tasks which can be treated as trainable parameters along with the parameters of CNN. In practice, instead of $\sigma_1^2, \sigma_2^2$, their log values, i.e., $\log \sigma_1^2, \log \sigma_2^2$, are optimized to improve numerical stability.





- fixed weight scheme with $w_1 = 100, w_2 = 1$ where fixed weight ratio $w_1/w_2$ is chosen based on preliminary results of dynamic weight scheme. See Sect. SM3 in supplementary materials for access details.

To investigate the effects of MTL, we additionally train the CNN in the STL mode by using only one branch of the head. For notational simplicity, we do not differentiate two weighting schemes hereafter and refer CNN trained by MTL with better overall performance on $\mathcal{D}_{\mathrm{val}}$ as MTDL models, CNN trained by STL as STDL models.

### 2.3.3 Loss function

Mean Squared Error loss (MSELoss) has been commonly selected as the loss function for the task of regression, which is defined as:

$$\mathrm{MSELoss}(\hat{y}, y) = (\hat{y} - y)^2 \tag{6}$$

where $\hat{y}$ is the predicted value and $y$ is the reference value.

However, MSE places much more emphasis on samples with large absolute residuals during the fitting process and may thus result in a far less robust estimator when applied to grossly mismeasured data (Hastie et al., 2009). Therefore, other metrics such as Mean Absolute Error (MAE) and their combinations are adopted for model training on noisy data. In this work, we use Adaptive Huber Loss (AHL) defined by (Huber, 1964):

$$\mathrm{HuberLoss}(\hat{y}, y) = \begin{cases} \dfrac{1}{2\delta}(\hat{y} - y)^2, & \text{if } |\hat{y} - y| < \delta \\ |\hat{y} - y| - \dfrac{\delta}{2}, & \text{Otherwise} \end{cases} \tag{7}$$

where $\delta$ is a threshold that represents the estimated regression residual caused by potential outliers.

AHL avoids some undesired optimization issues caused by the constant norm of the gradient of MAE and is also less sensitive to the outliers when compared with MSE. However, the choice of $\delta$ becomes another concern for its implementation. In this work, we initialize $\delta$ by the Normalized Median Absolute Deviation (NMAD) of the training dataset (Hastie et al., 2009) which is defined by

$$\mathrm{NMAD} = 1.4826 \times q_{50}\left(|y_j - q_{50}(y_j)|\right) \tag{8}$$

where $j$ is traversed from 1 to $N_{\mathrm{train}}$ and $q_{50}$ is the 50th percentile operator. Then $\delta$ is adjusted adaptively during the training process using the following equation

$$\delta^{(i)} = q_{90}\left(|\hat{y}_j - y_j|\right) \tag{9}$$

where $i$ is the number of finished epochs and $q_{90}$ is the 90th percentile operator.





### 2.3.4 Data augmentation

Multiple factors including light scattering mechanisms, atmospheric scattering conditions and equipped device health can bring non-negligible uncertainty and noise to the acquired satellite imagery, which presents a great challenge to the generalization ability of deep networks applied to a global scale (Zhu et al., 2017). While assembling adequate data can tackle this problem

from its root, it might be a daunting task due to the collection difficulty and computational cost. Instead of inflating the dataset directly, data augmentation uses data warping or oversampling during the training process to help deep networks efficiently learn certain translational invariances (Shorten and Khoshgoftaar, 2019). In this work, we employ the data warping strategy which randomly transforms the sample patch by flipping, color space shifting and noise injection while preserving the target value. To be specific, to mimic various noises during image acquisition, we add the Gaussian noise to Sentinel-1's VV, VH

bands and Sentinel-2's NIR band and the ISO noise, Gaussian noise and color jittering to Sentinel-2's red, green and blue bands, where transformation probabilities for each operation are all set to 0.5. Finally, after stacking all bands including DEM band, we randomly flip the multi-spectral sample patch with a probability of 0.5.

### 2.4 Technical implementation

The above designed DL models are implemented using PyTorch (https://pytorch.org/). To achieve faster convergence, we adopt

the cosine annealing with warm restarts (Loshchilov and Hutter, 2017) as our learning rate scheduler. To be specific, we choose the initial learning rate $\mathrm{lr}^{(0)}$ as 0.01 and then decrease it following a cosine curve:

$$\mathrm{lr}^{(i)} = \mathrm{lr}_{\min} + \frac{1}{2}\left(\mathrm{lr}^{(0)} - \mathrm{lr}_{\min}\right)\left(1 + \cos\left(\frac{T_{\mathrm{cur},t}}{T_t}\pi\right)\right) \tag{10}$$

where $\mathrm{lr}^{(i)}$ denotes the learning rate at the $i$-th epoch, $T_{\mathrm{cur},t}$ the number of epochs since the $t$-th restart and $T_t$ the number of epochs between the $t$-th and $t+1$-th restart. In this work, we trained CNN for 155 epochs with $T_0 = 5$, $T_1 = 10$, $T_2 = 20$,

$T_3 = 40$ and $T_4 = 80$.

For optimizer selection, we use Stochastic Gradient Descent (SGD) optimizer for single task learning and Adam optimizer for multi-task learning. Momentum factor of SGD optimizer is set to be 0.9. Running averages of gradient and its square of Adam optimizer are set to be 0.9 and 0.999, respectively. Weight decay factor of both optimizers is set to be 0.0001. Batch sizes used for gradient estimation is set to be 64 (256) when STL (MTL) is chosen. Pre-trained parameters are saved into PyTorch

**state_dict** objects which can be further loaded by **torch.nn.Module.load_state_dict** function.

The implemented pipeline has been published as an open source Python package, SHAFTS (Simultaneous building Height And FootprinT extraction from Sentinel Imagery), and its source code is available at https://github.com/LllC-mmd/3DBuildingInfoMap.

SHAFTS includes the following key functions:

– **sentinel1_download_by_extent**: download Sentinel-1's images from GEE according to the specified year, spatial extent and temporal aggregation.





- **`sentinel2_download_by_extent`**: download Sentinel-2's images from GEE according to the specified year, spatial extent and temporal aggregation.

- **`srtm_download_by_extent`**: download SRTM data from GEE according to the specified spatial extent.

- **`pred_height_from_tiff_DL_patch`**: predict building height and footprint using the STDL model.

- **`pred_height_from_tiff_DL_patch_MTL`**: predict building height and footprint using the MTDL model

Detailed usage of these functions refer to the homepage of SHAFTS (https://github.com/LllC-mmd/3DBuildingInfoMap).

## 3 Model evaluation

### 3.1 Baseline models and evaluation metrics

Based on the review of related work (Li et al., 2020; Frantz et al., 2021), we select RFR, SVR and its bagging version (BaggingSVR), XGBoostR (Chen and Guestrin, 2016) as ML baselines for comparison with DL models. See Appendix A for implementation details of ML approaches. Also, STDL models are developed as benchmarks to investigate the effects of MTL.

To give quantitative assessment to the performance of our developed models, experimental results are evaluated on $\mathcal{D}_{\text{test}}$ using multiple metrics including Root Mean Square Error (RMSE) and its weighted version (wRMSE), Mean Absolute Error

(MAE), Mean Error (ME), Normalized Median Absolute Deviation (NMAD), Peason correlation coefficient (CC), coefficient of determination ($\mathrm{R}^2$) and its weighted version ($\mathrm{wR}^2$) defined in Table 2.

Before we embark on the specific results of model evaluation, it is worth reflecting on the incentives for the selection of these metrics.

For the magnitude of model error, RMSE and MAE are common measures using $L^2$ and $L^1$ norm. Compared with MAE,

RMSE can be sensitive to the performance degradation of models caused by outliers such as skyscrapers which may be scarce in the real urban environment. Thus, Frantz et al. (2021) proposed to use RMSE weighted by the sample frequency (wRMSE) as a measure of the areal accuracy of models. Besides the magnitude, both center location and dispersion are key aspects of error distribution, which can be described by ME and NMAD, respectively. The reason why NMAD rather than the sample standard deviation is selected is that the former is considered to be a more robust estimator for the standard deviation than the

latter (Carrera-Hernández, 2021).

Furthermore, since the distribution of the target dataset may vary significantly across scales, some dimensionless normalized metrics are preferred for the convenience of inter-comparison in relative terms. CC is widely used as an overall skill score for assessing structural similarity (Li et al., 2020; Taylor, 2001). A higher CC suggests a more consistent tendency in variation between reference and prediction if any linear transformation is allowed. However, since linear transformation may discard the

information about the center location and amplitude of error distribution, $\mathrm{R}^2$ is further selected to provide those complementary information quantifying the overall model performance. Originally, $\mathrm{R}^2$ indicates the percentage of target variance explained by a linear regression model. It can be further extended to the nonlinear case to reflect the improvement over the benchmark





**Table 2.** Summary of metrics used in this work. $y, \hat{y}$ stand for reference value and predicted value, respectively. $\Delta y_i = \hat{y}_i - y_i$. $\mu_y$ is the sample mean of $y$. $\sigma_y, \sigma_{\hat{y}}$ are the sample standard deviations of $y$ and $\hat{y}$. $\sigma_{y,\hat{y}}$ is the sample covariance between $y$ and $\hat{y}$. $w_i$ is the weight of regression residual determined by the frequency of corresponding target value which can be estimated using Gaussian Kernel Density Estimation.

| Type | Metric | Definition |
|------|--------|-----------|
| Magnitude | RMSE | $\mathrm{RMSE} = \sqrt{\frac{1}{N_{\text{test}}} \sum_{i=1}^{N_{\text{test}}} (\Delta y_i)^2}$ |
| | wRMSE | $\mathrm{wRMSE} = \sqrt{\frac{1}{N_{\text{test}}} \sum_{i=1}^{N_{\text{test}}} w_i (\Delta y_i)^2}$ |
| | MAE | $\mathrm{MAE} = \frac{1}{N_{\text{test}}} \sum_{i=1}^{N_{\text{test}}} |\Delta y_i|$ |
| Distribution | ME | $\mathrm{ME} = \frac{1}{N_{\text{test}}} \sum_{i=1}^{N_{\text{test}}} \Delta y_i$ |
| | NMAD | $\mathrm{NMAD} = 1.4826 \times q_{50} \left( |\Delta y_i - q_{50}(\Delta y_i)| \right)$ |
| Skill scores | CC | $\mathrm{CC} = \frac{\sigma_{y,\hat{y}}}{\sigma_y \sigma_{\hat{y}}}$ |
| | $R^2$ | $R^2 = 1 - \frac{\sum_{i=1}^{N_{\text{test}}} (\Delta y_i)^2}{\sum_{i=1}^{N_{\text{test}}} (y_i - \mu_y)^2}$ |
| | $\mathrm{wR}^2$ | $\mathrm{wR}^2 = 1 - \frac{\sum_{i=1}^{N_{\text{test}}} w_i (\Delta y_i)^2}{\sum_{i=1}^{N_{\text{test}}} w_i (y_i - \mu_y)^2}$ |

model if mean target value is thought to be an appropriate choice (Schaefli and Gupta, 2007). Thus, for representation of urban morphology, $R^2$ can be interpreted as a skill metric which quantifies the performance improvement or degradation of models

in capturing surface heterogeneity relative to homogeneous approximation using the sample mean.

## 3.2   Patch-level evaluation of prediction skill

### 3.2.1   Performance overview

For building height prediction, as shown in Fig. 4, ML models achieve RMSE ranging from 9.46 m to 6.85 m, $R^2$ varying from 0.173 to 0.455 while DL models achieve RMSE ranging from 5.26 m to 3.95 m, $R^2$ varying from 0.718 to 0.818. The

performance gaps between ML models and DL models are narrowed if we use wRMSE and $\mathrm{wR}^2$ for overall assessment. But wRMSE measures are still reduced by 0.49 m to 1.76 m when comparing DL models with ML models, which is equivalent to nearly 27%-61% reduction in ML model's overall error. Thus, DL models show significant overall performance improvement over ML models. Furthermore, DL models give more robust estimations since DL models achieve much smaller error variance than ML models for all cases in terms of NMAD. Among ML models, RFR shows the poorest performance and is inclined

to over-smoothing which over-estimates low height value and under-estimates high height value. And SVR exhibits the best capability of height prediction among three ML models, which agrees with the result of previous research (Frantz et al., 2021).

For building footprint prediction, as shown in Fig. 5, ML models achieve $R^2$ ranging from 0.377 to 0.846 and DL models achieve $R^2$ varying from 0.834 to 0.972. Thus, when compared with building height prediction, overall performance gaps between ML models and DL models of building footprint prediction are much smaller. For most developed models except

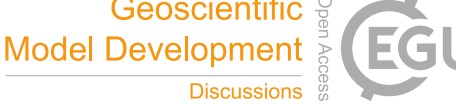

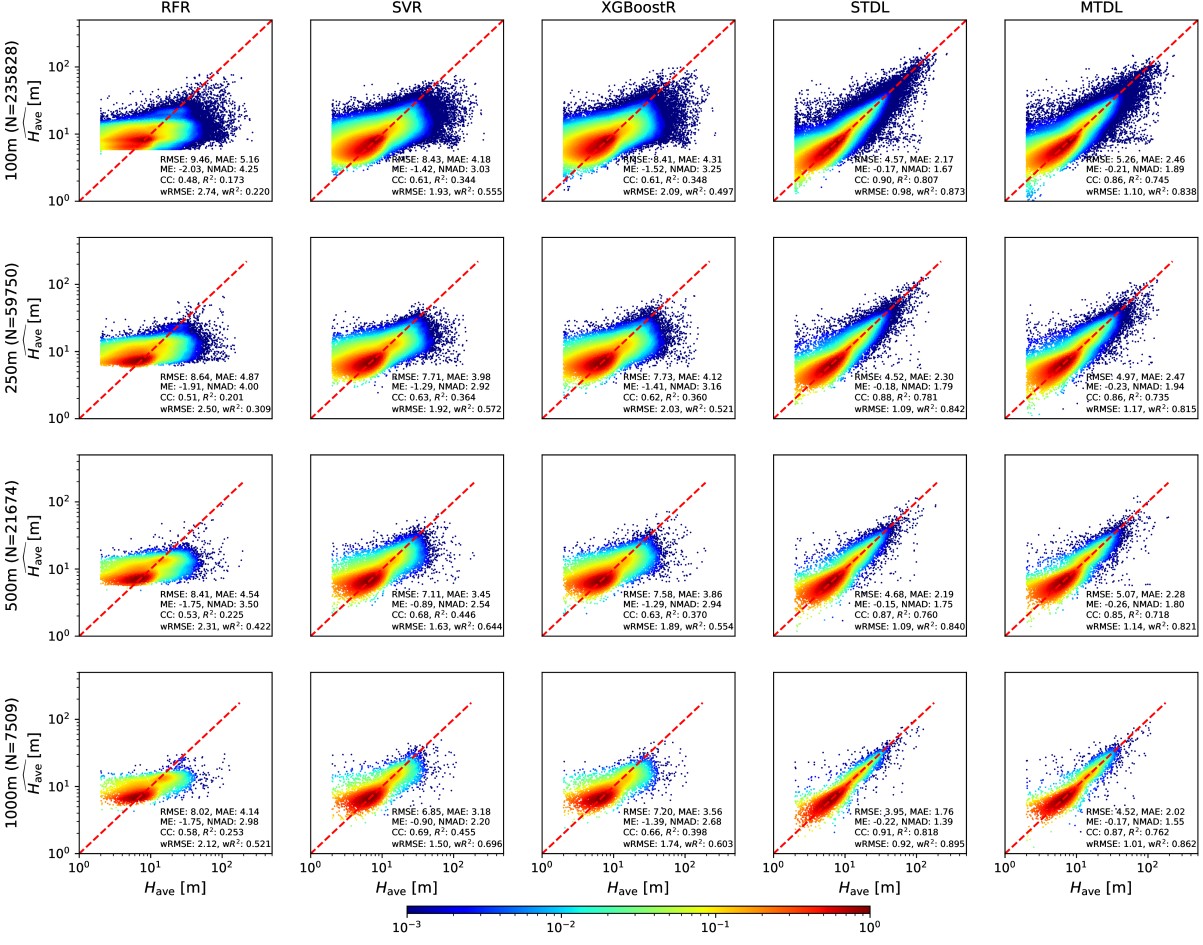

**Figure 4.** $H_{ave}$ predicted by ML models and DL models. The density of scatter points is normalized to $[10^{-3}, 1]$ and represented using shading colors.

for RFR, when the target resolution decreases, area with high density rotates gradually and results in a more symmetric and concentrated distribution on the left and right of the one-to-one line. This tendency of area with high density would make the slope closer to 1.0 when regressing the reference against the predicted values, which indicates the improvement of performance measured by the quotient of CC divided by $\sigma_{\hat{y}}/\sigma_y$ (Gupta et al., 2009).

### 3.2.2   Stratified error assessment

To further investigate the model performance in different target domains, we made stratified error assessment by calculating RMSE, ME and NMAD over the sub-datasets which are split from the original dataset $\mathcal{D}_{test,2}$ by the Cartesian product between two sets of intervals with respect to reference $H_{ave}$ and $\lambda_p$. The summary of error metrics is presented in Fig. 6 and Fig. 7.





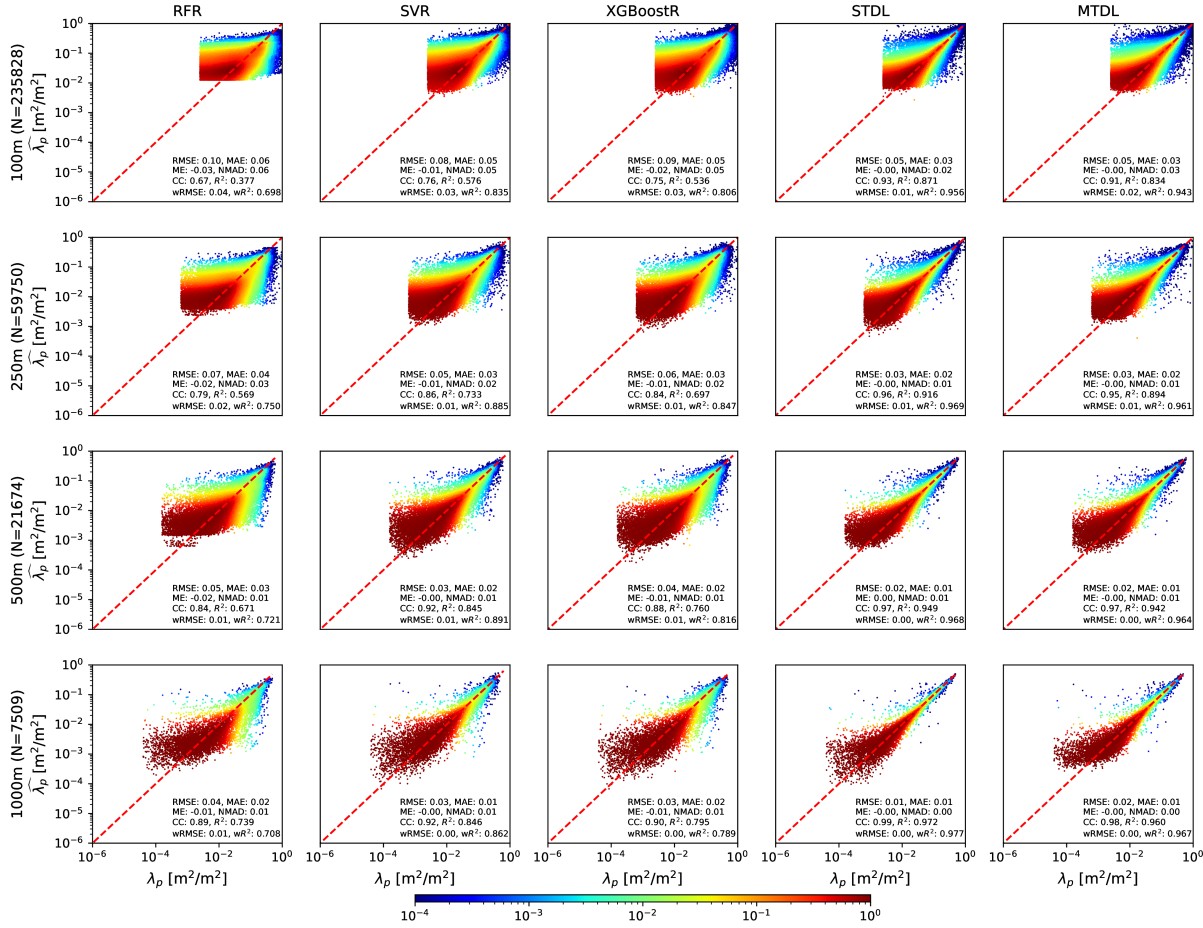

**Figure 5.** $\lambda_p$ predicted by ML models and DL models. The density of scatter points is normalized to $[10^{-3}, 1]$ and represented using shading colors.

For building height prediction in different target domains, SVR shows large systematic underestimation for $H_{ave}$ above 20 m in all cases, which is recognized as the saturation effect in previous research (Frantz et al., 2021; Koppel et al., 2017). In contrast
to SVR, both STDL and MTDL models exhibit greatly mitigated saturation effects for the prediction of $H_{ave}$ ranging from 20 m to 40 m, indicating the suitability of DL models for high-risers prediction. However, average systematic underestimation of $H_{ave}$ above 40 m can still be -18.3 m for STDL models, -28.5 m for MTDL models in the case of 100 m, which might result from the scarcity of samples whose average building height is higher than 40 m (cf. Fig. 2). Besides improving high-risers prediction, when compared with SVR, DL models also reduce the average systematic overestimation of building height lower
than 5 m by 42%-70% based on the statistics of all cases. Similar phenomenon of systematic underestimation exists when SVR makes predictions on target $\lambda_p$ larger than 0.25 where DL models achieve 61%-81% reduction on ME. Also, both SVR and DL models tend to overestimate the target $\lambda_p$ smaller than 0.05, especially in the case of 100 m. This might be largely attributed



**Figure 6.** Error metrics of building height prediction in different target domains. Only results from SVR are presented since it achieves the best performance among three ML models.

to the limited resolution of Sentinel's imagery where small buildings in low-density built-up area can not be clearly resolved. The problem of systematic overestimation is mitigated when target resolution becomes coarser since more information from the surrounding environment can be utilized to infer building existence under a medium resolution of 10 m.

Based on the preliminary results of stratified analysis on ME, we further grouped target domains of $H_{ave}$ and $\lambda_p$ into three representative intervals separately. And we summarized the average error magnitude measured by RMSE and standard deviation of error measured by NMAD over these intervals in Fig. 8. See Sect. SM4 in supplementary materials for detailed quantification results of model performance. Both SVR and DL models show relatively unfavorable RMSE and NMAD for the low-value domain but DL models behave slightly better. When it comes to medium-value domain where the majority of samples





**Figure 7.** Same as Fig. 6, but for building footprint prediction.

lie (cf. Fig. 2), for building height prediction, DL models reduce RMSE and NMAD by 2-3 m and 0.4-1.5 m, respectively, when compared with SVR. For building footprint prediction in the medium-value domain, performance improvement brought by DL models tends to increase with the coarsening of target resolution. Such a phenomenon indicates that building footprint prediction can get more benefit from the growing size of input patches when compared with building height prediction. For the high-value domain, although both error magnitude and standard deviation of error become much larger, it can be naturally attributed to the increasing dynamic range of target values (Geiß et al., 2019). Furthermore, when compared with SVR in the high-value domain, DL models reduce RMSE by 31%-50% for building height prediction and 40%-67% for building footprint



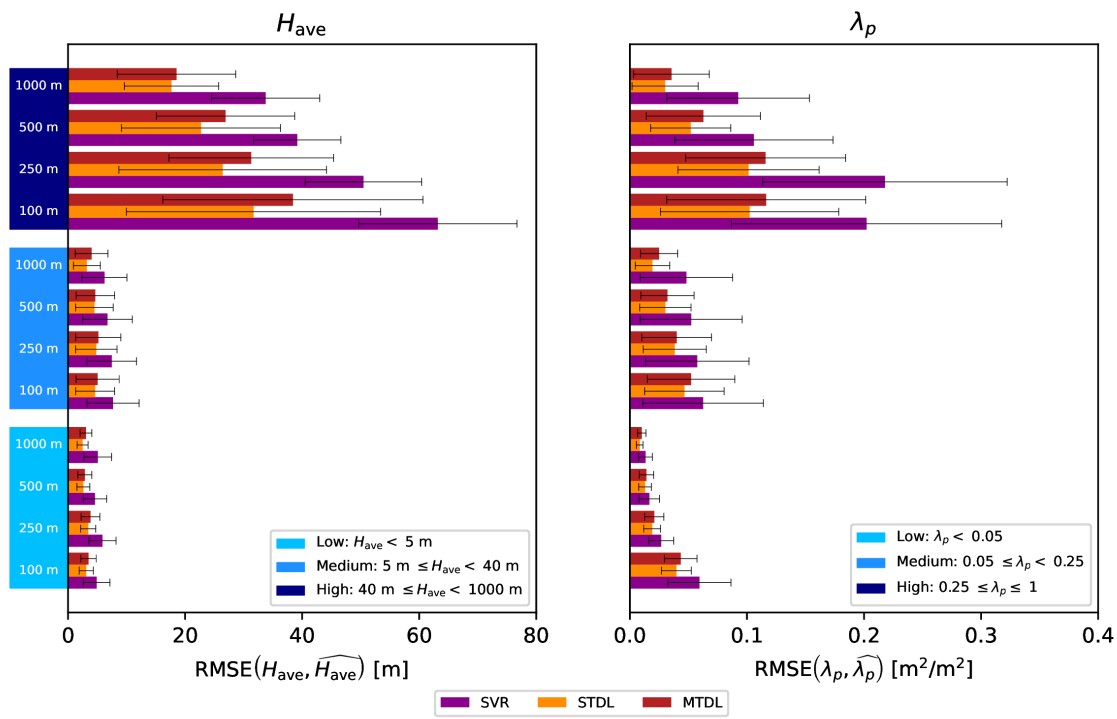

**Figure 8.** RMSE and NMAD of building height and footprint prediction over representative intervals where NMAD corresponds to half the length of error bars distributed symmetrically at the ends of boxes. For building height prediction, the low-value, medium-value and high-value domains are defined by $H_{ave} < 5$ m, $5$ m $\leq H_{ave} < 40$ m, $40$ m $\leq H_{ave} < 1000$ m, respectively. For building footprint prediction, the low-value, medium-value and high-value domains are defined by $\lambda_p < 0.05$, $0.05 \leq \lambda_p < 0.25$, $0.25 \leq \lambda_p \leq 1$, respectively.

prediction, which demonstrates the superiority of DL models on the 3D building information prediction in target domains with high $H_{ave}$ and $\lambda_p$.

Furthermore, Fig. 7 reveals the dependence between error metrics of building footprint prediction and target building height: larger error magnitude and uncertainty of building footprint prediction can be noticed for domains with higher building height. It might be caused by the layover and shadow effect brought by high-risers on surrounding environment which hinders models from capturing all existing building footprints precisely (Stilla et al., 2003).

### 3.2.3 Diagnosing performance gaps between DL and ML models

To further investigate why the DL models outperform the ML ones, we conducted a decomposition analysis of $R^2$ (Fig. 9).

Intuitively, an appropriate prediction requires both magnitude and variability of the underlying distribution to be captured accurately. Following this insight, a causal link can be drawn among proposed error metrics to promote the model evaluation towards a more diagnostic sense. $1 - R^2$ can be related with RMSE using normalization (Gupta et al., 2009) and then decom-





posed into the summation of the magnitude part measured by ME and the variability part measured by centered RMSE (Taylor,

425 2001)

$$1 - \mathrm{R}^2 = \frac{\sum_{i=1}^{N_{\text{test}}} (y_i - \hat{y}_i)^2}{\sum_{i=1}^{N_{\text{test}}} (y_i - \mu_y)^2} = \left(\frac{\mathrm{RMSE}}{\sigma_y}\right)^2 = \left(\frac{\mathrm{ME}}{\sigma_y}\right)^2 + \left(\frac{\mathrm{RMSE}'}{\sigma_y}\right)^2$$

$$= \mathrm{ME}_n^2 + \mathrm{RMSE}'^2_n$$

(11)

where $\mathrm{RMSE}'$ is the centered RMSE defined by

$$\mathrm{RMSE}' \triangleq \left(\frac{\mathrm{RMSE}'}{\sigma_y}\right)^2 = \sqrt{\frac{1}{N_{\text{test}}} \sum_{i=1}^{N_{\text{test}}} [(\hat{y}_i - \mu_{\hat{y}}) - (y_i - \mu_y)]^2}$$

(12)

where $\mu_{\hat{y}}$ is the sample mean of $\hat{y}$.

Basically, Eq. 11 indicates that variation in model performance measured by $\mathrm{R}^2$ (or RMSE) consists of systematic deviation from center quantified by ME and associated variability measured by $\mathrm{RMSE}'$. Since systematic deviation represented by ME is easier to be corrected, more emphasis can be laid on the analysis of the variability error measured by $\mathrm{RMSE}'$.

Taylor (2001) gives a decomposition of $\mathrm{RMSE}'$ following the form of the law of cosines

$$\mathrm{RMSE}'^2_n = \left(\frac{\mathrm{RMSE}'}{\sigma_y}\right)^2 = \left(\frac{\sigma_{\hat{y}}}{\sigma_y}\right)^2 + 1 - 2 \cdot \frac{\sigma_{\hat{y}}}{\sigma_y} \cdot \mathrm{CC}$$

(13)

Eq. 13 suggests that $\mathrm{RMSE}'$ is influenced by variability consistency quantified by CC and variability magnitude quantified by $\sigma_{\hat{y}}$. The similarity between this decomposition formula and the law of cosines facilitates the model comparison by a single intuitive image called Taylor diagram (Taylor, 2001).

According to Fig. 9, we first decompose $1 - \mathrm{R}^2$ into the summation of a systematic bias $\mathrm{ME}_n$ and a variability error $\mathrm{RMSE}'_n$. And we find the variability error (i.e., $\mathrm{RMSE}'^2_n$) dominates the variation of $1 - \mathrm{R}^2$ (Fig. 10) and is about 50% smaller in DL

models than in ML ones. Furthermore, $\mathrm{RMSE}'^2_n$ of building height and footprint prediction mainly shows a decreasing trend when target resolution becomes coarser, which indicates the improvement of model skills on capturing variability. For all cases, $\mathrm{ME}_n^2$ only accounts for a small fraction of $1 - \mathrm{R}^2$: it can be largely ignored in DL models but stays at a quasi-constant non-negligible level in ML ones, especially for building height prediction. Such decomposition suggests that, when compared with ML models, the significant improvement brought by DL models are mainly due to considerable reduction in variability errors.

However, it is also remarkable that for ML models, owing to almost invariant level of systematic bias, it can be expected to occupy a much larger percentage of total error when target resolution becomes coarser.

Since the variability error dominates the overall bias, we further attribute the variation of $\mathrm{RMSE}'^2_n$ to the magnitude part measured by $\sigma_{\hat{y}}/\sigma_y$ and the consistency part measured by CC (Fig. 11).

For building height prediction, CC of ML models mainly vary between 0.48 and 0.69, which indicates the moderate corre-

lation between reference and prediction. And DL models further improve the results of CC by 0.17 to 0.42. Besides, a larger





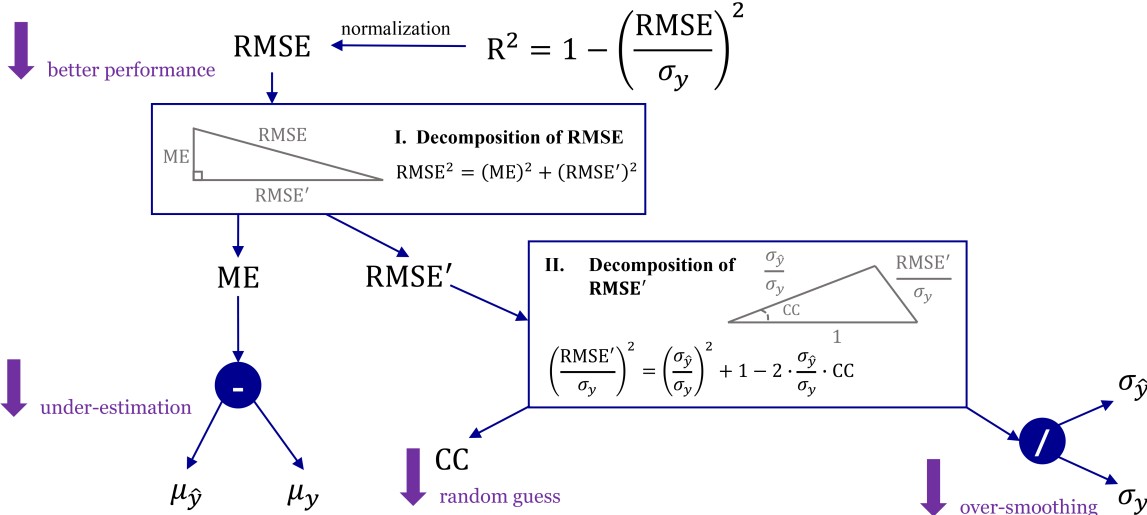

**Figure 9.** Decomposition of RMSE and $R^2$ where two boxes represent two steps of decomposition analysis. Colored arrows indicate the computation flow for error metrics and colored circles represent the computation operators where "-" denotes $\mu_{\hat{y}} - \mu_y$ and "/" denotes $\mu_{\hat{y}}/\mu_y$. See Table. 2 for detailed definitions of symbols.

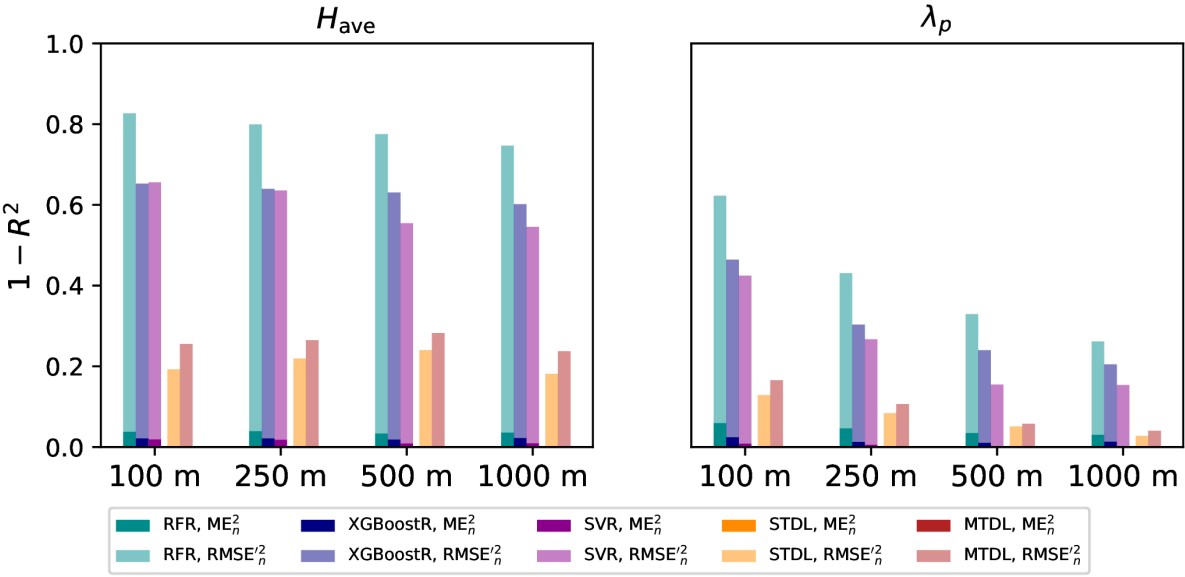

**Figure 10.** Decomposition of $1 - R^2$ for model predictions.

magnitude (0.22–0.56) of difference in $\sigma_{\hat{y}}/\sigma_y$ can be found between ML and DL models. Therefore, the better performance of DL models over ML ones can be mainly attributed to the improvement of variability magnitude.



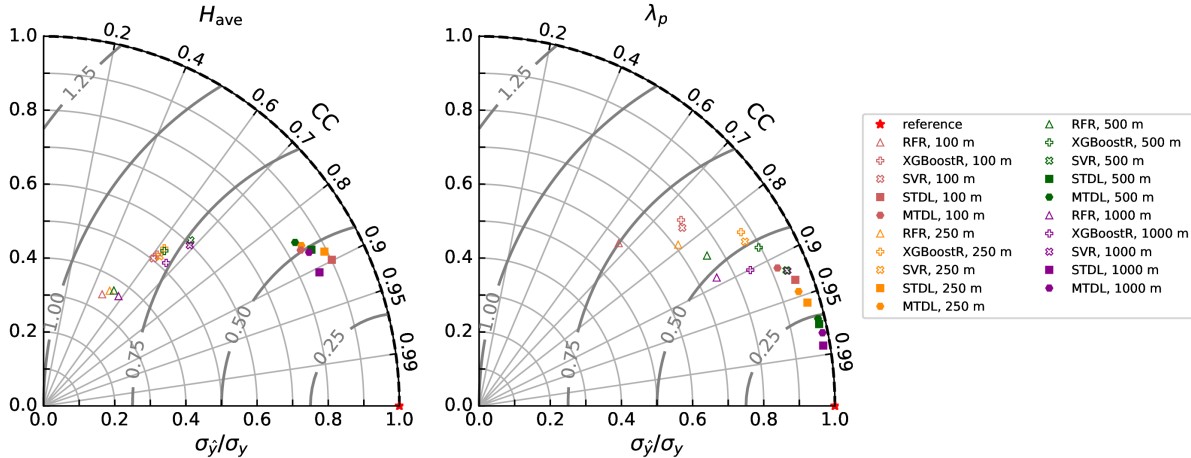

**Figure 11.** Taylor plots of comparing model performance across different scales where contours represent different levels of $\text{RMSE}'_n$.

However, for building footprint prediction, the variation of model performance seems to be more complicated. This phenomenon might result from the mixture effects of model and dataset. Reference height distribution shows minimal variability when the resolution changes from 100 m to 1000 m (Fig. 2) and thus largely reduces the influence caused by distribution transformation. However, improvement brought by DL models in both CC and $\sigma_{\hat{y}}/\sigma_y$ can still manifest (Fig. 11) though the magnitude of improvement becomes less remarkable when compared with building height prediction. The narrowed performance difference between DL and ML models also agrees with the comparison results with respect to overall performance on $H_{\text{ave}}$ and $\lambda_p$ (Sect. 3.2.1).

### 3.2.4 The effects of MTL

For patch-level evaluation, STDL models achieves better performance than MTDL ones (Fig. 10 and Fig. 11). However, the performance gap between STDL and MTDL models narrows when target resolution becomes coarser (Fig. 10). This is probably due to more contextual information gradually ingested by larger input patches. When target resolution decreases from 100 m to 1000 m, input size increases from 200 m to 1600 m and thus more contextual information can be utilized to extract representative features. This can be further evidenced by t-SNE visualization (van der Maaten and Hinton, 2011) of features extracted by the backbone part of DL models (Fig. 12): for building height prediction, when compared with manually constructed features for ML models, automatically learned features of DL models make sample points with different magnitude of height values gradually separated into several groups when target resolution decreases, indicating features with more discriminative power are learned by DL models.

Compared with STDL models, although MTDL models only achieve comparable performance (consistent with a previous pixel-level result of Cao and Huang (2021)), MTDL models are more efficient as evidenced by nearly halving number of parameters compared to STDL ones (Table. 3). This conclusion can be further confirmed by the t-SNE visualization of features





**Figure 12.** Visualization of features for building height prediciton in $\mathcal{D}_{\text{test},2}$ using t-SNE. Features of STDL and MTDL models are defined as the output from the backbone part while features of ML models are extracted using techniques mentioned by Sect. A1.

for building footprint prediction (Fig. 13). Transformed distribution of features extracted by MTDL models remains same for both cases whereas boundaries splitting different types of sample points are still maintained clearly. In contrast, features
learned by STDL models exhibit a totally different and more mixed form of distribution compared with the case of building height prediction. Since both STDL and MTDL models learn the representation of the same dataset, much redundancy can be expected in the learning results of STDL models.




**Table 3.** Summation of number of parameters for simultaneous building height and footprint prediction using STDL and MTDL models under different target resolutions.

| Target resolution | 100 m | 250 m | 500 m | 1000 m |
|---|---|---|---|---|
| STDL | 2734242 | 2766242 | 2766242 | 10950818 |
| MTDL | 1418962 | 1434962 | 1434962 | 5681490 |

**Figure 13.** Visualization of features for building footprint prediciton in $\mathcal{D}_{\text{test},2}$ using t-SNE.



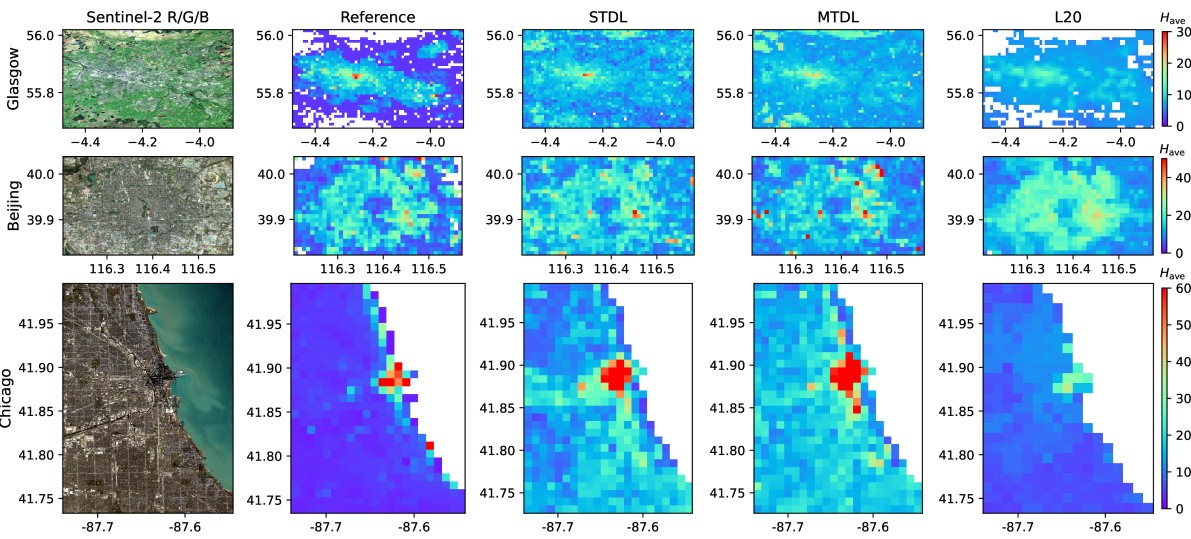

**Figure 14.** Building height close-ups in Glasgow, Beijing and Chicago.

## 3.3    City-level evaluation of spatial transferability

### 3.3.1    Qualitative evaluation in Glasgow, Beijing and Chicago

To further investigate the spatial transferability of our developed DL models, we mapped 3D building structures at 1000 m resolution for three metropolitan area from different continents, namely, Glasgow, Beijing and Chicago (Fig. 14 and Fig. 15). As the height information of Beijing and Chicago is only available as floor numbers, it is converted into building height with an assumed storey height of 3 m following Cao and Huang (2021); Li et al. (2020). Also, we compare our results against a recent 3D mapping product based on RFR (Li et al. (2020); L20 hereinafter), which utilizes multi-source data including remote

sensing imagery and additional information including urban footprint mask and roads (Cao and Huang, 2021; Frantz et al., 2021).

As mentioned before, publicly available reference data inevitably introduce noises and such noisy reference data pose a challenge to model performance evaluation, especially in terms of quantitative pixel-wise comparison (Burke et al., 2021). For example, after preliminary assessment of reference data quality in three cities, we found that for Chicago, although most

building polygons exhibit satisfactory agreement with building footprints interpreted from aerial images, there are about 48% missing entries in the height field of building polygons, which may cause significant deviation between calculated reference and underlying truth. Thus, we only evaluate mapping results of Beijing and Chicago in a qualitative way.

For building height prediction, DL models can better capture peak values, especially for skyscrapers neighboring Lake Michigan in Chicago (Fig. 14). However, overestimates are observed for a large area of Chicago and Beijing when compared

with reference maps. While it agrees with the findings of overestimation in low-value domains based on previous patch-level analysis (Fig. 6), it might also be caused by the bias in reference building height converted from the number of floors when



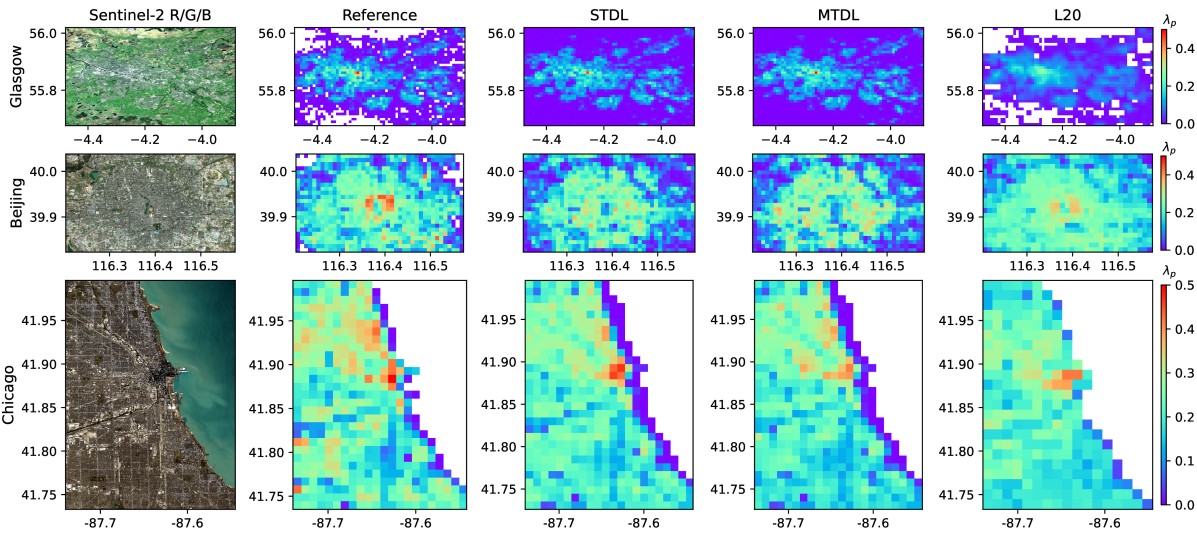

**Figure 15.** Same as Fig. 14, but for building footprint.

compared with their actual values. Based on existing records of some iconic buildings and street view images given by Baidu map (https://map.baidu.com/) and Google Maps (https://www.google.com/maps/), we find that deriving building height from the unit of floor numbers using a constant ratio of 3.0 may underestimate actual reference building height, especially for high-

risers. Take China Zun in Beijing for example, it has an actual height of 528 m but only 108 floors, which results in a 38.6% underestimate if a height/floor ratio of 3.0 is used in the conversion. Thus, more advanced strategies would be desired if we aim to harmonize datasets from these two units for model development and evaluation.

     Compared with building height, reference data of building footprint is expected to be more reliable based on the aforementioned data quality assessment. For building footprint prediction, all of three models can give a relatively accurate estimation

of the urban boundary and spatial distribution of building footprint. But L20 tends to provide more blurred results such as the mapping for the south area of Beijing due to overestimation in building footprint. This over-smoothing phenomenon might be caused by the prediction strategy of RFR where final result was determined by a simple average of results from all trees (Hastie et al., 2009; Liu et al., 2021). Furthermore, all of three models are likely to underestimate the peak values of high-density built-up area, especially in Chicago, where the MTDL model gives slightly better predictions on these areas.

To investigate possible reasons why underestimation exists, we looked into the Sentinel-2's optical images of these areas: some buildings can be hardly identified precisely even by human eyes due to coarse resolution and misleading reflectance (not shown). Thus, besides Sentinel imagery, images with finer resolution or additional urban information such as multi-source datasets used in L20 may help improve the building footprint prediction.





**Table 4.** Building height and footprint prediction evaluated in Glasgow. Error metrics including RMSE, $\mathrm{ME}_n^2$ and CC are calculated using pixels from the intersection part of masked reference and predictions.

| Metric | | RMSE | | | $\mathrm{ME}_n^2$ | | | $\mathrm{RMSE'}_n^2$ | | | CC | | |
|---|---|---|---|---|---|---|---|---|---|---|---|---|---|
| Model | | STDL | MTDL | L20 | STDL | MTDL | L20 | STDL | MTDL | L20 | STDL | MTDL | L20 |
| $H_{\mathrm{ave}}$ | 100 m | 3.76 | 3.31 | / | 0.02 | 0.00 | / | 0.70 | 0.56 | / | 0.64 | 0.67 | / |
| | 250 m | 3.12 | 2.88 | / | 0.01 | 0.00 | / | 0.57 | 0.49 | / | 0.70 | 0.71 | / |
| | 500 m | 2.97 | 2.74 | / | 0.01 | 0.00 | / | 0.57 | 0.50 | / | 0.67 | 0.71 | / |
| | 1000 m | 2.61 | 2.47 | 2.96 | 0.00 | 0.00 | 0.09 | 0.53 | 0.48 | 0.61 | 0.69 | 0.73 | 0.65 |
| $\lambda_p$ | 100 m | 0.08 | 0.09 | / | 0.00 | 0.01 | / | 0.48 | 0.51 | / | 0.74 | 0.71 | / |
| | 250 m | 0.05 | 0.05 | / | 0.01 | 0.01 | / | 0.44 | 0.44 | / | 0.75 | 0.75 | / |
| | 500 m | 0.06 | 0.06 | / | 0.00 | 0.02 | / | 0.81 | 0.78 | / | 0.54 | 0.55 | / |
| | 1000 m | 0.04 | 0.04 | 0.06 | 0.02 | 0.01 | 0.35 | 0.49 | 0.51 | 0.71 | 0.72 | 0.72 | 0.57 |

### 3.3.2 Quantitative evaluation in Glasgow

To perform quantitative evaluation under the challenge of noisy data, we focus our evaluation on urbanized areas with more reliable building morphological information. Thus, we implemented a post-processing algorithm for urban boundary extraction, the city clustering algorithm based on percolation theory (CCAP) (Cao et al., 2020), which utilizes the mapping results of building height and footprint to automatically determine an optimal threshold of the footprint value differentiating between urban and non-urban area (see Appendix B for details). In the following parts, we present Glasgow's masked mappings and
evaluation results after performing CCAP.

**3D building structure mapping**

The results of comparing our mappings and L20 with reference data are summarized in Table. 4.

When compared with L20 benefited from multi-source datasets, although our 1000 m mappings are merely based on Sentinel imagery, they still achieve 16.6% and 33.3% improvement on RMSE for building height and footprint estimation, respectively.
For building height prediction, although L20 achieves a relatively close CC with DL models, it has much higher $\mathrm{ME}_n^2$ and $\mathrm{RMSE'}_n^2$ than DL models and therefore suffers from overall performance degradation. A similar but much larger systematic bias can be observed for building footprint prediction when comparing L20 with the MTDL model. Thus, it can be concluded that systematic bias would greatly influence the performance of RFR at a relatively coarser resolution such as 1000 m, especially for building footprint mappings, which agrees with previous statistical results derived by the decomposition of $1 - \mathrm{R}^2$ (Sect.
3.2.3).

Moreover, we compare the performance with respect to STDL and MTDL models. For building height prediction, when compared with STDL models, MTDL models achieve better performance in all cases, especially for the case of 100 m where the MTDL model reduces the RMSE by 0.45 m. From the decomposition part of RMSE at the case of 100 m, this can be mainly





attributed to the decrease of $\mathrm{RMSE}'^2_n$, which indicates that the MTDL model captures the variability of building height more
correctly than the STDL model when target resolution becomes refined. This phenomenon seems contradictory to previous
patch-level results and our intuitions that multi-task learning might hurt the performance as the size of input patches decreases
because less information is available for correlating building height with footprint. Nevertheless, since multi-task learning
necessitates a trade-off for a single model by joint optimization of two prediction tasks and further constrains the parameter
space of desired models (Sener and Koltun, 2018), MTDL models can be expected to have better generalization ability and
reduced risk of fitting noises. This might help us understand why MTDL models can exhibit better performance of building
height prediction in the spatial transferability testing at a refined scale such as 100 m. For building footprint prediction, STDL
and MTDL models achieve almost same performance in terms of both RMSE and its decomposed parts.

Since above error metrics assess model performance in an average sense, we further plotted the predicted spatial distribution
of building height and footprint in Fig. 16 and Fig. 17, respectively, with medians of corresponding bias in different target
domains. For both building height and footprint mappings, our models can clearly resolve central urban area and correspond-
ing peak values. However, for all cases, urban area identified from predictions are larger than those derived from reference,
especially for the 100 m case. It is mainly caused by the fact that a large number of external pixels lying outside the central area
have small height values due to missing data so that they would always be identified as non-urban pixels during the process of
CCAP. No significant difference can be found when we compare spatial distributions of $H_{\mathrm{ave}}$ and $\lambda_p$ predicted by STDL and
MTDL models. However, plots of medians of bias in Fig. 16 and Fig. 17 reveal that although STDL and MTDL models still
exhibit relatively small performance gaps for all target domains of $H_{\mathrm{ave}}$, they mainly differ in the domain where $H_{\mathrm{ave}} \geq 21$ m,
especially for cases of 100 m and 1000 m. In that domain of the case of 100 m, the STDL model shows a much smaller absolute
value of the median of bias for the case of 100 m. This phenomenon seems to disagree with previous results obtained by the
comparison of RMSE. However, it should be noted that stratified bias analysis neglects sample frequency of target domains
and therefore can lead to its disagreement with error metrics derived by averaging of overall samples.

Moreover, stratified bias analysis also gives new insights into the comparison between L20 and proposed DL models. For
building height prediction, although L20 achieves a slightly smaller absolute value of the median of bias in the domain where
$H_{\mathrm{ave}} < 10$ m, the performance gap between L20 and proposed DL models is reversed and then enlarged up to 10 m in the do-
main where $24$ m $\leq H_{\mathrm{ave}} < 27$ m. A similar performance gap up to 0.15 can also be observed for building footprint prediction
ranging from 0.4 to 0.5. Thus, it can be concluded that multi-source information still fails to effectively compensate for the
weakness of ML models in predicting high-risers and high-density built-up area and therefore promoting advances in model
architecture can be expected to bring more benefits.

**Uncertainty analysis of input Sentinel-1's band**

To further investigate potential effects caused by radiometric performance degradation of Sentinel-1's images on 3D building
information mapping, we performed uncertainty analysis on prediction results by injecting random noises to the input VV
and VH bands. Specifically, according to absolute radiometric accuracy for the IW mode reported in the official Sentinel-
1 annual performance report for 2020, we designed three levels of zero-mean Gaussian noises with standard deviation $\sigma =$
$0.1, 0.3, 0.5$ dB. And for each level of noises and target resolution, we performed 50 random simulations by adding independent



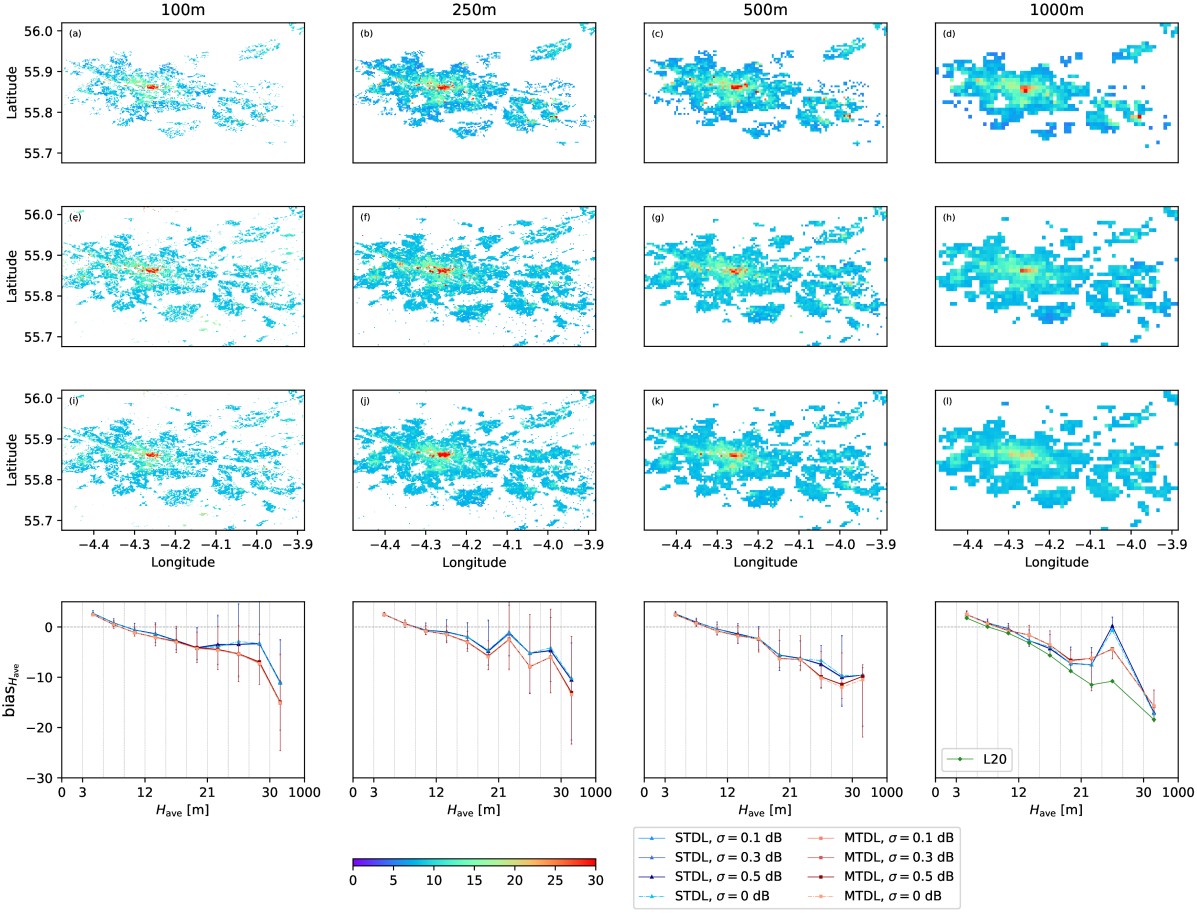

**Figure 16.** Building height prediction in Glasgow with corresponding stratified bias distribution. (a)-(d): reference building height maps. (e)-(h): building height maps predicted by STDL models. (i)-(l): building height maps predicted by MTDL models. The bias is equal to prediction minus reference. $\sigma$ is the standard deviation of Gaussian noises injected to the Sentinel-1's VV and VH bands. $\sigma = 0$ dB stands for the results using original Sentinel's images with no noise injected. Uncertainty in each target domains is expressed as a vertical line whose upper, lower bounds and middle points represent the quartiles of bias distribution. Target domains of building height are divided by: 3 m, 6 m, 9 m, 12 m, 15 m, 18 m, 21 m, 24 m, 27 m, 30 m. Domains without valid samples are omitted.

noises to original VV and VH bands. Fig. 16 and Fig. 17 show quartiles of bias distribution over different target domains

derived from random simulations. By comparing quartiles of stratified bias under different levels of radiometric noises, no significant variation can be observed when the noise level increases from 0.1 dB to 0.5 dB, which demonstrates the robustness of DL models with respect to internal noises with $\sigma < 0.5$ dB. Furthermore, when comparing STDL with MTDL models, the approximate invariance of stratified bias distributions can lead to similar conclusions drawn from original uncorrupted images with respect to the median of bias. However, for building height prediction, relatively more minor uncertainty of bias measured

by IQR is presented by MTDL models in the high-value domain such as $H_{ave} \geq 24$ m in the case of 100 m. It can be naturally



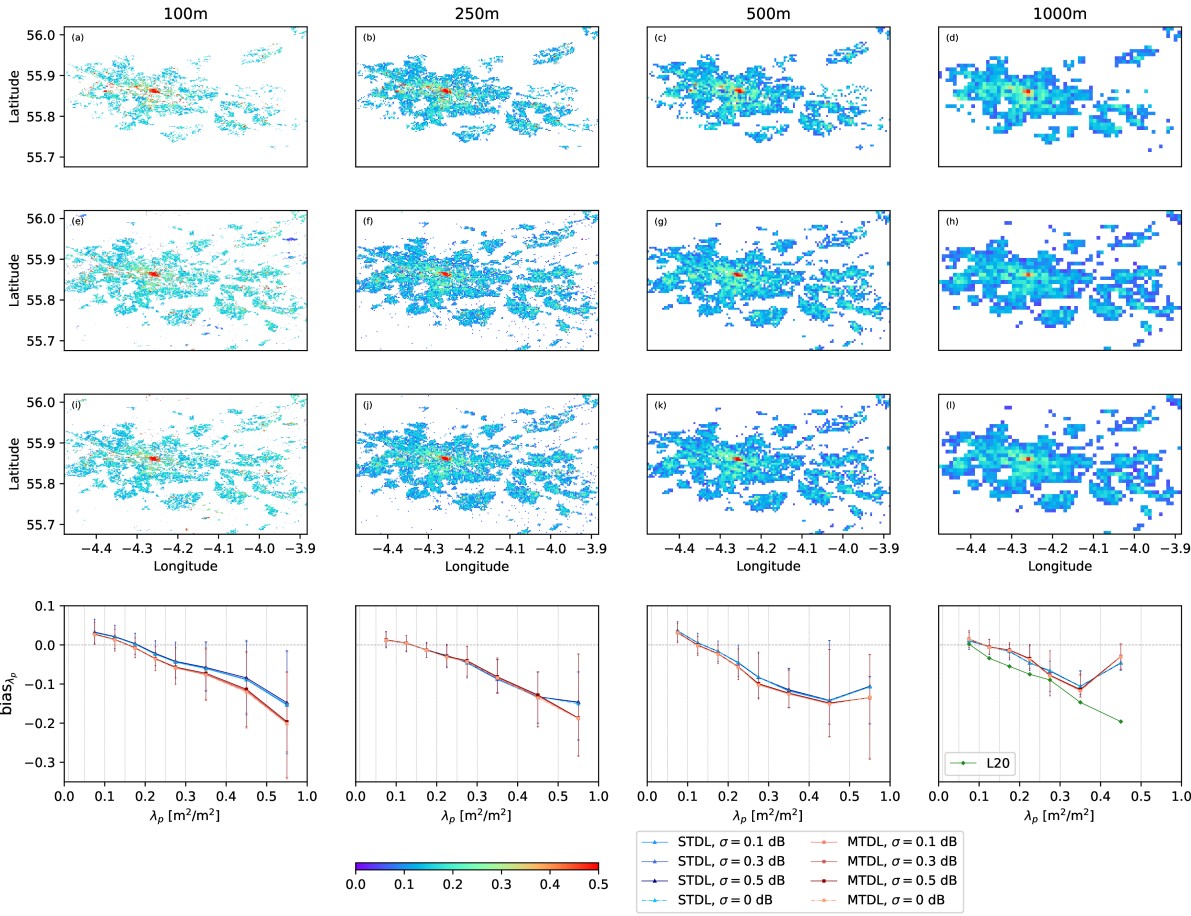

**Figure 17.** Same as Fig. 16, but for building footprint and corresponding target domains are divided by: 0.001, 0.01, 0.05, 0.1, 0.15, 0.20, 0.25, 0.3, 0.4, 0.5.

attributed to the aforementioned regularization effect merited from multi-task learning. Thus, MTDL models can be promising in reducing the uncertainty of peak value prediction as well as increasing overall accuracy (cf. Table. 4) for the refined building height mapping.

## 4    Conclusions

In this work, we develop a MTDL Python package SHAFTS for the simultaneous extraction of both building height and footprint from Sentinel imagery in urban areas. Comparison with conventional ML and STDL models in 46 cities worldwide demonstrates that:



1. DL models - both STDL and MTDL - avoid tedious feature engineering by automatic data representation and outperform their ML counterparts in building height and footprint prediction for 38 cities by 0.27-0.63, 0.11-0.49 measured by $R^2$, respectively. Particularly, DL models effectively mitigate the saturation effect of ML models in high-value domains: for building height prediction, RMSE is reduced by 31 m when target height is higher than 40 m; for building footprint prediction, RMSE is reduced by 0.1 when target footprint is larger than 0.25.

2. Within the DL family, MTDL models achieve higher efficiency than STDL models by halving the number of required parameters and exhibit better generalization ability (e.g., in Glasgow with an improvement of 0.45 m in building height prediction measured by RMSE as well as reduced uncertainty of bias for the prediction of building height higher than 24 m for the case of 100 m).

Furthermore, we establish a diagnostic evaluation framework based on a two-step decomposition analysis, which can better interpret the performance variation across different models and target resolutions. Using this framework, we find that DL models outperform ML models in both systematic bias control and variability error reduction. In addition, relatively large systematic bias may significantly deteriorate the performance of ML models at coarse resolutions (e.g., > 1000 m) and high-value domains.

Being promising for 3D building mapping in a large scale, the MTDL models in SHAFTS have several limitations that need to be appreciated here. Overestimation on building height and underestimation on building footprint of the developed MTDL model is observed in some densely built-up areas, which could be caused by the limited spatial resolution of Sentinel imagery and caveats in the designed models. Also, the inherent noises in reference data may influence the accuracy in assessing model performance. Therefore, robustness of the MTDL model will be tested in more cities in our forthcoming work. Moreover, the fusion of multi-source input data and more advanced ensemble strategies between ML and DL models will be conducted to address these issues and improve the prediction of urban morphological features.

*Code and data availability.* The snapshot of source codes and reference datasets has been archived as a zipped file on Zenodo (https://doi.org/10.5281/zenodo.6587510) to guarantee the reproducibility of our research. And the up-to-date version of SHAFTS is available at: https://github.com/LllC-mmd/3DBuildingInfoMap. SHAFTS is also accessible via the Python `pip` package manager.

## Appendix A: ML approaches

### A1 Feature extraction and preprocessing

To prepare the inputs for ML models, following (Geiß et al., 2020), 396 features are manually designed and extracted from sample patches in three aspects (see Table. A1 for summary):

- Single-band statistic: the mean, standard deviation, maximum, minimum, 25th percentile, 50th percentile and 75th percentile of SRTM DEM, Sentinel-1's VV, VH bands and Sentinel-2's red, green, blue and NIR bands;



**Table A1.** Features extracted from Sentinel-1 and Sentinel-2 imagery as the inputs of ML models. More details on feature description and calculation methods can be found in Geiß et al. (2020); Haralick et al. (1973); Pesaresi and Benediktsson (2001).

| Category | Feature |
|---|---|
| single-band statistic | $VV_{mean,std,max,min,q_{25},q_{50},q_{75}}$ |
| | $VH_{mean,std,max,min,q_{25},q_{50},q_{75}}$ |
| | $DEM_{mean,std,max,min,q_{25},q_{50},q_{75}}$ |
| | $Red_{mean,std,max,min,q_{25},q_{50},q_{75}}$ |
| | $Green_{mean,std,max,min,q_{25},q_{50},q_{75}}$ |
| | $Blue_{mean,std,max,min,q_{25},q_{50},q_{75}}$ |
| | $NIR_{mean,std,max,min,q_{25},q_{50},q_{75}}$ |
| multi-spectral indices | $n(NIR\text{-}Red)_{mean}$ |
| | $n(NIR\text{-}Green)_{mean}$ |
| | $n(NIR\text{-}Blue)_{mean}$ |
| | $n(NIR\text{-}Green)_{mean}$ |
| | $Brightness_{mean}$ |
| morphological characteristic | $DMP \text{ based on Opening}_{mean}^{R,G,B,NIR,Brightness}$ |
| | $DMP \text{ based on Closing}_{mean}^{R,G,B,NIR,Brightness}$ |
| | $DMP \text{ based on Opening-By-Reconstruction}_{mean}^{R,G,B,NIR,Brightness}$ |
| | $DMP \text{ based on Closing-By-Reconstruction}_{mean}^{R,G,B,NIR,Brightness}$ |
| texture metrics | $GLCM \text{ mean}_{mean}^{R,G,B,NIR,Brightness}$ |
| | $GLCM \text{ variance}_{mean}^{R,G,B,NIR,Brightness}$ |
| | $GLCM \text{ homogeneity}_{mean}^{R,G,B,NIR,Brightness}$ |
| | $GLCM \text{ contrast}_{mean}^{R,G,B,NIR,Brightness}$ |
| | $GLCM \text{ dissimilarity}_{mean}^{R,G,B,NIR,Brightness}$ |
| | $GLCM \text{ entropy}_{mean}^{R,G,B,NIR,Brightness}$ |
| | $GLCM \text{ angular 2nd moment}_{mean}^{R,G,B,NIR,Brightness}$ |
| | $GLCM \text{ correlation}_{mean}^{R,G,B,NIR,Brightness}$ |

    – Multi-spectral information: six normalized difference spectral indices and a brightness measure (Zhang et al., 2017).

    – Morphological and texture quantities: differential morphological profiles (DMP) and gray-level co-occurrence matrix (GLCM) calculated using a square-shaped structuring element with six ascending sizes $\mathcal{S} = \{3, 5, 7, 9, 11, 13\}$. Based on DMP and GLCM, several morphological and texture features were derived using different aggregation functions.

To reduce information redundancy and irrelevant noises in high-dimensional data, we preprocess features in the following steps:






1. Filtering: raw features with variance lower than $1e^{-6}$ are removed from the input feature set.

2. Normalization: remaining features are standardized by removing individual means and scaling to unit variance.

3. Transformation: principal component analysis is performed on normalized features to project them onto a lower dimensional space with a reduction ratio of 50%.

**A2    ML models**

This work selects RFR, XGBoostR, SVR and its bagging version (BaggingSVR) as baseline models for comparison with DL
models. It should be pointed out that while accuracy is our major concern for model selection, scalability is another crucial
factor if the model would be applied to real-world situations for massive data mining. Compared with RFR and XGBoostR,
most existing implementation of SVR with non-linear kernels exhibits poorer scalability since it involves the solution of a
quadratic optimization problem, which often requires well-designed decomposition algorithms (Joachims, 1998). Thus, when
predicting on the dataset with the resolution of 100 m and 250 m, the original SVR is replaced with a bagging regressor
(Breiman, 1996), i.e., BaggingSVR, which trains SVR on random subsets of the original dataset as base estimators and then
makes the final prediction with the aggregation of individual results. For notational simplicity, we denote both of them as SVR.

To guarantee the reliability of our research, we use widely used open-source Python packages for the implementation of ML
models: **scikit-learn** (Varoquaux et al., 2015) for preprocessing pipelines, RFR, SVR and **XGBoost** (Chen and Guestrin,
2016) for XGBoostR (see key hyperparameter values in Table. A2).

**Appendix B: the city clustering algorithm based on percolation theory (CCAP)**

The procedure of CCAP can be summarized into the following steps:

1. Construct a potential building footprint threshold set with uniform spacings

$$\mathcal{S}_{\lambda_p} = \left\{ \lambda_{p_i} \mid \lambda_{p_{i+1}} - \lambda_{p_i} = \Delta\lambda_p, i = 1, ..., N-1 \right\} \tag{B1}$$

2. Determine an optimal footprint threshold $\lambda_{p_i}^*$ based on percolation theory (Cao et al., 2020)

$$\lambda_{p_i}^* = \underset{i}{\mathrm{argmax}}\,\mathrm{Entropy}(\lambda_{p_i}) \tag{B2}$$

where $\mathrm{Entropy}(\cdot)$ is the Shannon's entropy of the size distribution of the urban cluster system. Here a pixel is identified as an
urban pixel if its building height and footprint value $(\lambda_p, H_{\mathrm{ave}})$ satisfies $\lambda_p \geq \lambda_{p_i}, H_{\mathrm{ave}} \geq 5.0\mathrm{m}$ and urban clusters are extracted
by the Connected Component Analysis algorithm provided in the **OpenCV** package (https://github.com/opencv/opencv).

3. Determine the threshold for final masking. Since different optimal thresholds can be calculated from the mapping results
of various models (e.g., STDL and MTDL models), final masking can be derived by some simple fusion rules. In this work,
we consider a pixel as an urban pixel if it is identified so in any mapping result.



**Table A2.** Hyperparameters optimization for ML models. For BaggingSVR, we choose n_svr=40 and max_samples=0.05 for the case of 100 m, n_svr=50 and max_samples=0.1 for the case of 250 m.

| Model | Hyperparameter name | Description | Parameter value |
|---|---|---|---|
| RFR | n_tree | the number of trees in the forest | 500 |
| | max_depth | the maximum depth of the tree | 9 |
| SVR | C | regularization parameter for the L2 penalty | 1.0 |
| | epsilon | the width of epsilon-tube within which no penalty is applied | 0.1 |
| BaggingSVR | n_svr | the number of base SVR estimators in the ensemble | 30, 50 |
| | max_samples | the percentage of samples to draw from the whole dataset to train each base SVR estimator | 0.05, 0.1 |
| XGBoostR | n_estimators | number of gradient boosted trees | 500 |
| | max_depth | maximum tree depth for base learners | 9 |
| | reg_lambda | regularization parameter for the L2 penalty | 1.0 |
| | gamma | minimum loss reduction required to make a further partition on a leaf node of the tree | 0.1 |

*Author contributions.* RL led the development of SHAFTS. RL and TS performed the formal analysis. TS, GN and FT contributed to the computing resources and supervision. RL wrote the original draft and all authors contributed to review and editing of the paper.

*Competing interests.* The authors declare that they have no conflict of interest.

*Acknowledgements.* This work was supported by the National Key Research and Development Program of China (2018YFA0606002), Open Research Fund Program of State Key Laboratory of Hydroscience and Engineering (sklhse-2020-A06), NERC Independent Research Fellowship (NE/P018637/1).



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
