# Peer review of "SHAFTS (v2022.3): a deep-learning-based Python package for Simultaneous extraction of building Height And FootprinT from Sentinel Imagery"

_Geoscientific Model Development, 2022_

## Author Comment (AC2)

We thank the reviewers and editor for their review of our paper and appreciate the helpful suggestions, which are incorporated in the revised manuscript.

Responses refer to the new Section/Figure/Table/Appendix in the revised manuscript unless otherwise indicated. Related revisions are excerpted in salmon with grey background with page (P) and line (L) numbers where appropriate.

**Reviewer 1**

**General Comments:**

*This study develops a deep-learning (DL) based Python package-SHAFTS to extract 3D building information (average building height and footprint) from publicly available satellite imagery. Compared to conventional machine learning-based models and single-task DL models, the proposed multi-task DL models can effectively improve the prediction accuracy. This study involves the fusion of multi-source input data and many machine learning and deep learning models, which undoubtedly requires huge and solid work from the authors. Although I am not the expert in computer science, the evaluation framework presented in Section 3 is scientifically sound from my perspective – very quantitative from patch-level to city level. And I will consider using the developed package in the future. I only have the following minor comments.*

Thank you for your constructive comments. We have carefully considered your suggestions and incorporated them in the revised manuscript.

**Specific Comments:**

1. *Line 368, the caption of Figure 4, need to denote the source of reference values when calculating RMSE, MAE, etc.*

We have revised the caption of Figure 4 per your suggestions as follows (P18):

> Figure 4. $H_{ave}$ predicted by ML models and DL models. The density of scatter points is normalized to $[10^{-3}, 1]$ and represented using shading colors. The presented evaluation metrics are calculated based on the reference values from $\mathcal{D}_{test,2}$ which consists of samples from 43 cities by patch-level sampling (See Sect. 2.2.3 for details).

2. *Figure 5, is the density of scatter points normalized to [10-4, 1] instead of [10-3, 1]?*

Corrected as suggested. The revised caption of the Figure 5 now reads as follows (P19):

> Figure 5. $\lambda_p$ predicted by ML models and DL models. The density of scatter points is normalized to $[10^{-4}, 1]$ and represented using shading colors. The presented evaluation metrics are calculated based on the reference values from $\mathcal{D}_{test,2}$ which consists of samples from 43 cities by patch-level sampling (See Sect. 2.2.3 for details).

3. *Line 385, this paragraph shall better describe the stratified error, which is helpful to the readers who are not familiar with it.*

Thank you for your helpful comment. We have rewritten the paragraph as follows (P18L385–P18L387):

> To further investigate the model performance in certain target domains, we made stratified error assessment in two steps. We first split the original dataset $\mathcal{D}_{test,2}$ into combinations of different subsets of reference $H_{ave}$ and $\lambda_p$. And then we calculated RMSE, ME and NMAD over each sub-dataset. The summary of error metrics is presented in Fig. 6 and Fig. 7.

4. *Line 404-405, "Both SVR and DL models show relatively unfavourable RMSE and NMAD for the low-value domain but DL models behave slightly better." What does it mean? I think the RMSE and NMAD are both smaller in the low-value domain (Fig. 8).*

We thank the reviewer for pointing out this confusing statement – the wording around "unfavourable" needs to be interpreted with the consideration of typical values of the prediction variables within the stratified domains, which, however, we missed to include in the previous manuscript. Using average building height at the resolution of 100 m by MTDL models as an example, although its RMSE in the low-value ($H_{ave} < 5$ m) domain is lower than that of the medium-value domain (5 m $\leq H_{ave} < 40$ m) (3.50 m vs. 5.05 m), the corresponding relative error – RMSE normalized by the median – of the former domain is remarkably larger than the latter one (140.0% vs. 22.4%). Thus, we indicated that relatively "unfavourable" model performance may be observed for the low-value domain.

We have now clarified this point in the revised manuscript (P20L404–P21L407):

> Both SVR and DL models show lower RMSE and NMAD for the low-value domain compared with other domains, which, however, should be interpreted with the consideration of data ranges of respective domains – relatively unfavorable model performance may be observed for the low-value domain if quantified with RMSE/NMAD normalized by domain-specific reference values (e.g. medians).

5. *Line 463, what is contextual information?*

The contextual information refers to the statistical properties of the surrounding environment – such as the spatial distribution of surface elevation, land cover and building shadows – can benefit perceptual inference tasks (Mottaghi et al., 2014). We have incorporated the related explanation and reference into the revised manuscript (P25L465–P25L468):

> When target resolution decreases from 100 m to 1000 m, input size increases from 200 m to 1600 m and thus more contextual information about the surrounding environment – such as the spatial distribution of surface elevation, land cover and building shadows – can be utilized to extract features, which can benefit downstream inference tasks (Mottaghi et al., 2014).

6. *12 and 13, why the feature patterns of STDL and MTDL have large differences?*

The large differences in feature patterns between STDL and MTDL models are due to the different number of branches used in the representation learning – STDL models use one branch of the heads whereas MTDL models use two branches. This has been clarified in the revised manuscript as follows (P26L473–P26L476):

> Compared with STDL models, although MTDL models exhibit different learned feature patterns due to the fact that they are trained using two branches of the heads (See Sect. 2.3.2 for details), they can achieve comparable performance (consistent with a previous pixel-level result of Cao and Huang (2021)). Also, MTDL models are more efficient as evidenced by nearly halving number of parameters compared to STDL ones (Table. 3).

7. *Line 509, I think STDL model gives better predictions than others from Fig. 15?*

We thank the reviewer for pointing out this confusing statement which may result from the unclear definition of "these areas".

To clarify our rationale for the better model performance of MTDL, we performed a more detailed comparison between MTDL and STDL by looking into two areas A and B in the high-density built-up region of Chicago using connected component analysis on the threshold-filtered building footprint map:

area A (outlined by red lines in Fig. R1) is determined by the boundary of the maximum connected high-density pixels with an average building footprint value higher than the upper quantile of the entire region ($\sim 0.30$); while area B (outlined by black dot lines in Fig. R1) is a sub-region of A which contains pixels with $\lambda_p > 0.5$.

[Figure]

**Figure R1.** Close-ups of building footprint for Chicago with high-density built-up areas outlined by red lines and surrounding areas of peak values outlined by black dot lines.

For the area A, the MTDL model gives comparable predictions with the STDL model with comparable RMSE = $\sim 0.049$ for both models. While for area B, although the STDL model gives better predictions on the central peak values, it tends to overestimate the average built-up area. In contrast, the MTDL model achieves a more satisfactory balance between the high and low values. Also, the quantitative evaluation in area B indicates that MTDL slightly outperforms STDL with respect to RMSE (0.061 vs. 0.056). Therefore, we conclude that, compared with the STDL model, the MTDL one gives better predictions in the surrounding areas of peak values in the high-density built-up areas of Chicago.

We have incorporated the related clarification into the revised manuscript (P28L512–P30L517):

> Furthermore, all of three models are likely to underestimate the peak values of high-density built-up areas, especially in Chicago. In the high-density built-up areas in Chicago (outlined by red lines in Fig. 15), STDL and MTDL models achieve comparable performance: the STDL model exhibits better performance on the peak values but tends to overestimate the average built-up area in the corresponding surrounding region (outlined by black dot lines in Fig. 15); while for the MTDL model, it achieves a more satisfactory balance between the high and low values and gives slightly better predictions on these areas with respect to RMSE (0.061 for STDL vs. 0.056 for MTDL).

[Figure]

**Figure 15.** Same as Fig. 14, but for building footprint. In Chicago, high-density built-up areas and surrounding areas of peak values are outlined by red lines and black dot lines, respectively, where the area A is determined based on the connected component analysis on the map of pixels with an average building footprint value higher than 0.3 and the area B contains pixels with $\lambda_p > 0.5$.

**Reviewer 2**

**General Comments:**

*This study develops a deep-learning-based (DL) Python package–SHAFTS to extract average building height and footprint proportion at a pixelated level. I briefly read the paper, and think it is well-written and solid. I, therefore, believe it deserves publication.*

We appreciate your recognition of our work.

**Reference**

Mottaghi, R., Chen, X., Liu, X., Cho, N. G., Lee, S. W., Fidler, S., Urtasun, R., Yuille, A. (2014). The role of context for object detection and semantic segmentation in the wild. In *Proceedings of the IEEE conference on Computer Vision and Pattern Recognition*.